# Prostaglandin $E_2$ controls the metabolic adaptation of T cells to the intestinal microenvironment

Matteo Villa [1,2] ✉, David E. Sanin [1,3], Petya Apostolova [1,3,4], Mauro Corrado [1,5,6,7], Agnieszka M. Kabat[1,3], Carmine Cristinzio[1,8], Annamaria Regina [1,9], Gustavo E. Carrizo[1], Nisha Rana[1], Michal A. Stanczak [1], Francesc Baixauli [1], Katarzyna M. Grzes[1], Jovana Cupovic[1], Francesca Solagna[1], Alexandra Hackl[1], Anna-Maria Globig [10], Fabian Hässler [1], Daniel J. Puleston[1], Beth Kelly[1], Nina Cabezas-Wallscheid[1], Peter Hasselblatt[10], Bertram Bengsch [10,11], Robert Zeiser [4,11], Sagar [10], Joerg M. Buescher [1], Edward J. Pearce[1,3,11,12,13] & Erika L. Pearce [1,3,11,14] ✉

Immune cells must adapt to different environments during the course of an immune response. Here we study the adaptation of CD8+ T cells to the intestinal microenvironment and how this process shapes the establishment of the CD8+ T cell pool. CD8+ T cells progressively remodel their transcriptome and surface phenotype as they enter the gut wall, and downregulate expression of mitochondrial genes. Human and mouse intestinal CD8+ T cells have reduced mitochondrial mass, but maintain a viable energy balance to sustain their function. We find that the intestinal microenvironment is rich in prostaglandin $E_2$ ($PGE_2$), which drives mitochondrial depolarization in CD8+ T cells. Consequently, these cells engage autophagy to clear depolarized mitochondria, and enhance glutathione synthesis to scavenge reactive oxygen species (ROS) that result from mitochondrial depolarization. Impairing $PGE_2$ sensing promotes CD8+ T cell accumulation in the gut, while tampering with autophagy and glutathione negatively impacts the T cell pool. Thus, a $PGE_2$-autophagy-glutathione axis defines the metabolic adaptation of CD8+ T cells to the intestinal microenvironment, to ultimately influence the T cell pool.

The immune response is dynamic. Immune cells are activated, migrate, and perform effector functions in tissues characterized by different chemical and physical properties. In peripheral organs, immune cells surveil for pathogens as well as contribute to tissue homeostasis. At barrier sites such as the gut, immune cells are key to maintaining the symbiosis between host and microbiota, the breach of which promotes the development of inflammatory bowel disease (IBD). Understanding the mechanisms that control the adaptation of immune cells to the tissue niche is an important step toward identifying strategies to

modulate immune function and designing mucosal vaccines against infections and cancer[1].

To investigate how immune cells adapt to their microenvironment, we explored the intestinal CD8+ T cell response[2,3]. Upon activation in response to foreign antigens from the gut, naive CD8+ T cells ($T_N$) in the mesenteric lymph nodes (mLN) differentiate into effector/effector memory T cells ($T_{EM}$) or central memory T cells ($T_{CM}$). Differentiation into memory T cells is accompanied by transcriptional and metabolic reprogramming, which in turn underlie the characteristic

features of these cells[4–7]. $T_{EM}$ leave the lymph nodes and enter the blood stream to reach target organs (reviewed in ref. [8]). There, after clearing the antigen that initiated the immune response, only a small proportion of $T_{EM}$ survives, and continues to circulate in and out of peripheral tissues. Moreover, during the early phases of the immune response, a fraction of CD8+ T cells commits to the tissue-resident memory T cell ($T_{RM}$) fate, eventually seeding the gut tissue[9,10]. These long-lived and sessile cells constitute the front-line defense against future encounters to the same antigen[11–14]. The cytokines interleukin-15 (IL-15) and tissue growth factor-β (TGF-β), together with a network of transcription factors, coordinate the development of $T_{RM}$[15–21].

It is reasonable to hypothesize that adaptation of T cells to the gut environment influences the establishment of the intestinal CD8+ T cell pool. Indeed, some of the metabolic requirements of $T_{RM}$ are not shared across all tissues of residence, highlighting that tissue-specificity could impact metabolism[22,23]. Moreover, the surrounding microenvironment has been shown to influence the fate and function of CD8+ T cells. For instance, nutrient availability as well as cell-intrinsic metabolic programs impinge on fitness and function of tumor-infiltrating CD8+ T cells[24–27]. Similarly, tissue-specific cues regulate the transcriptional landscape and surface phenotype of T cells, thus influencing their residency[28,29]. These examples highlight how a variety of factors control the ability of T cells to adapt to and thrive in the surrounding environment.

Using single-cell RNA sequencing and oligo-tagged antibody staining, we resolved the transcriptional and metabolic adaptation of CD8+ T cells to the intestinal microenvironment. Upon sensing prostaglandin $E_2$ (PGE$_2$) in the gut tissue, CD8+ T cells decreased their mitochondrial content. PGE$_2$ induced mitochondrial depolarization in CD8+ T cells, and as a consequence, CD8+ T cells enhanced their anti-oxidant capability and engaged autophagy to clear mitochondria. Our findings further the knowledge of how the intestinal CD8+ T cell pool is established, define a precise metabolic adaption of T cells in a tissue, and highlight therapeutic avenues to modulate immune function in a tissue-restricted fashion.

## Results

### CD8+ T cells in the intestinal LP and IEL compartments reduce the expression of mitochondria-related genes

To study the tissue adaptation of CD8+ T cells, we focused on cells in the small intestine under homeostatic conditions. By combining cell sorting, single-cell RNA sequencing, and oligo-tagged antibody sequencing, we analyzed the transcriptome and surface proteome of CD8+ T cells from the mesenteric lymph node (mLN), lamina propria (LP) and intraepithelial lymphocyte fraction (IEL, concentrating on CD8αβ induced IEL) of the gut (Fig. 1a and Supplementary Fig. 1a), presuming that we would capture the transition of these cells across these sites under homeostatic conditions. To assess the tissue-resident nature of LP-isolated cells we used in vivo intravascular staining and found that CD8+ T cells isolated from the gut LP were virtually free of cells associated with blood vessels[30] (Supplementary Fig. 1b). Integrating the RNA and surface protein landscapes of 49,111 cells across 3 biological replicates we identified 10 clusters visualized on the uniform manifold approximation and projection (UMAP)-reduced dimensional space (Fig. 1b). Cell clusters were present in all biological replicates (Supplementary Fig. 1c) and displayed unique transcriptional and protein expression profiles (Supplementary Fig. 1d, e). In accordance with our sorting strategy, cluster 6 expressed Sell, Klf2, and S1pr1 (Supplementary Fig. 1d), which control retention of CD8+ T cells in lymphoid organs[31], and CD62L, pointing to their residence in mLN (Supplementary Fig. 1e). Expression of Rora, previously associated with intestinal residency[32], was observed in clusters 1 and 2, which also displayed increased expression of the ATP-to-AMP converting enzymes CD38 and CD39 (Supplementary Fig. 1d, e)[33]. Granzyme-encoding genes Gzma and Gzmb expression was increased in cluster 0

(Supplementary Fig. 1d), as well as CD103 and Integrin-β7, that help the positioning of CD8+ T cells in the gut (Supplementary Fig. 1e). Cells associated with clusters 0 and 5 also expressed CD326, better known as EpCAM, recently shown to be associated with IEL origin, to support the movement of T cells within the epithelial layer (Supplementary Fig. 1e[34]). Finally, expression of Lgals1 hinted at the ability of cells within cluster 7 to interact with the extracellular matrix (Supplementary Fig. 1d). These results highlighted the diversity of CD8+ T populations in the sites studied. Exploration of the tissue origin of the samples placed $T_N$ and $T_{CM}$ in close proximity to mLN $T_{EM}$ on the UMAP, whereas LP-isolated CD8+ T cells were between mLN $T_{EM}$ and IEL-isolated CD8+ T cells (Fig. 1c). Few clusters were shared between tissues, with cluster 6 almost exclusively populated by $T_N$ and $T_{CM}$ mLN cells; clusters 1, 2, and 4 were characteristic of LP, and clusters 0 and 5 originated mostly from the IEL fraction (Fig. 1d). A notable exception was cluster 7, which was distributed across both mLN and LP, suggesting a potential seeding point of mLN-activated CD8+ T cells in the intestinal LP (Fig. 1d). Altogether, we identified a substantial heterogeneity in the transcriptional landscape and surface profile of CD8+ T cells across the mLN-LP-IEL axis.

The single-cell analysis highlighted that CD8+ T cell adaptation to the LP and epithelial layer of the intestine correlated with remarkable changes in the expression of mitochondria-encoded genes (Fig. 1e). To better explore how these changes were regulated across the clusters identified in Fig. 1b, we ranked CD8+ T cells during their transition from the mLN to the gut according to transcriptional similarity using Slingshot[35]. We calculated the average pseudotime estimation for each cell based on all predicted trajectories, setting cells in cluster 6 as the origin of the progression based on our cell sorting strategy. In this manner, we ordered cells based on estimated pseudotime (Supplementary Fig. 2a). Accordingly, clusters in the LP and IEL appeared later in this progression and were characterized by a higher pseudotime value (Supplementary Fig. 2a). When we plotted the expression profiles of several mitochondria-encoded transcripts as a function of pseudotime, we found that CD8+ T cells progressively decreased their expression during the transition from mLN to LP and ultimately to IEL (Fig. 1f, Supplementary Fig. 2b). Of note, the progressive reduction of gene expression across pseudotime was not a generalized phenomenon, since a number of genes relevant to CD8+ T cell physiology followed different expression patterns during the transition from mLN to intestinal LP and epithelium (Supplementary Fig. 2c). In conclusion, CD8+ T cells transiting from the mLN to the intestine exhibited reduced expression of mitochondria-encoded transcripts.

### Intestinal LP- and IEL-isolated CD8+ T cells have reduced mitochondrial content

To further explore the results we acquired from our single cell sequencing analyses, we obtained blood and intestinal biopsies from healthy human donors and isolated CD8+ T cells (Supplementary Fig. 3a). We used flow cytometry to assess mitochondrial content in $T_N$, $T_{EM}$, CD45RA-expressing $T_{EM}$ ($T_{EMRA}$), and $T_{CM}$ isolated from the blood and $T_{RM}$ from the gut, by labeling with the mitochondria-selective dye Mitotracker green. $T_{RM}$ isolated from different areas of the gut showed a substantially reduced mitochondrial content as compared to blood-resident CD8+ T cell populations, recapitulating of our findings, in humans (Fig. 2a).

To further investigate the relationship between the adaptation of CD8+ T cells to the intestinal tissue and the reduction of mitochondrial mass, we explored the physiological CD8+ T cell gut immune response in naive mice (Supplementary Fig. 3b, further details about the gating strategy and the nomenclature used throughout the manuscript is in the "FACS" paragraph of the Methods section). Supporting the single cell and human data, we found that CD8+ T cells progressively reduced their mitochondrial content during the transition from mLN-resident subsets ($T_N$, $T_{EM}$, $T_{CM}$ and $T_{RM}$) to the LP and IEL, highlighting an

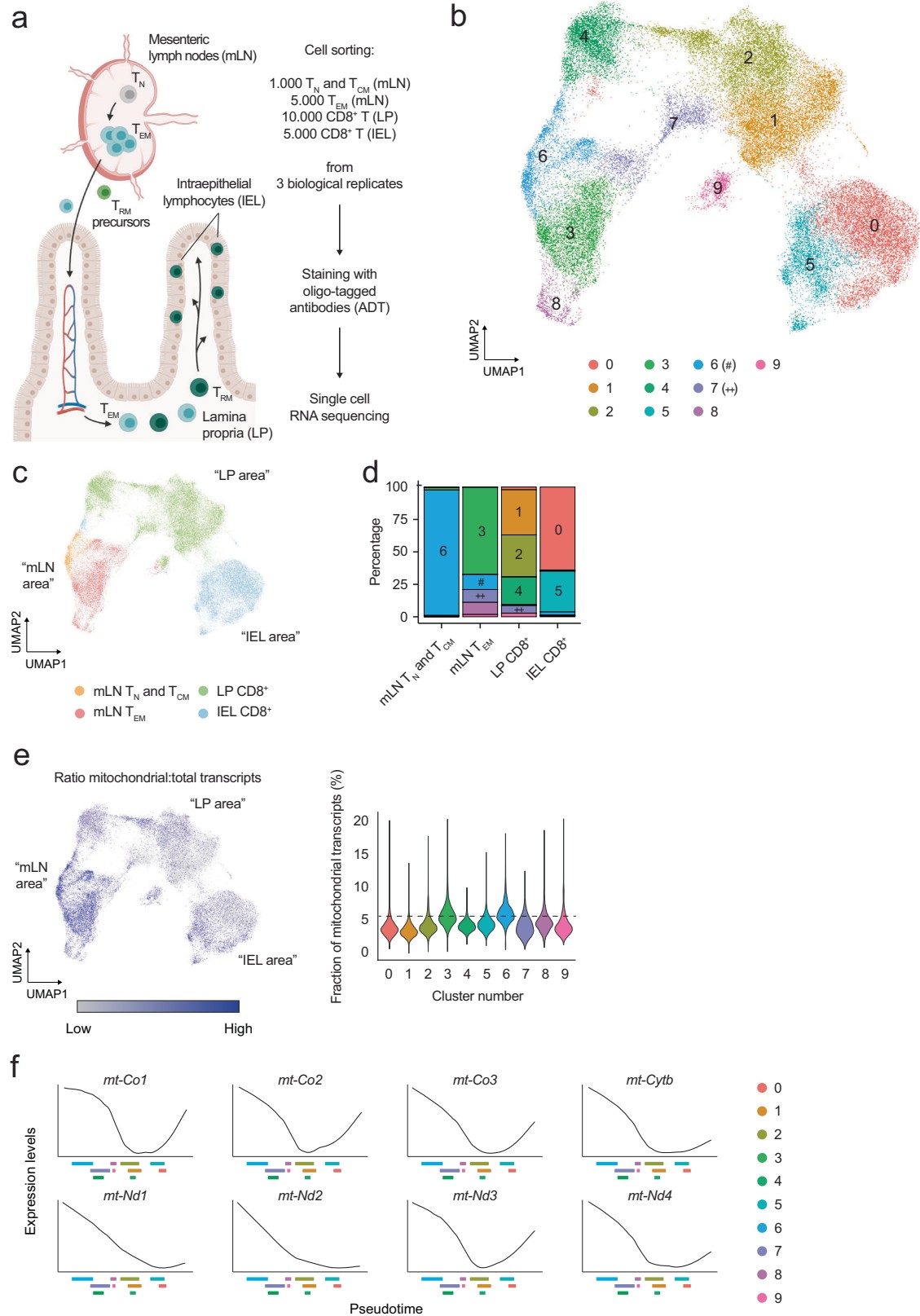

inflection point between the CD69⁻CD103⁻ and the CD69⁺CD103⁺ phenotypes in the LP (Fig. 2b). Honing in on the LP, and comparing CD69⁻CD103⁻ cells, CD69⁺CD103⁻ cells, and CD69⁺CD103⁺ cells[19], it became apparent that the reduction in mitochondrial content positively correlated with the CD69⁺CD103⁺ profile (Supplementary Fig. 3c). Of note, the reduction in mitochondrial content upon

adaptation to the LP and epithelial layer, as measured by Mitotracker green staining, was independent of the differential activity of mitochondrial efflux pumps (treatment with verapamil, a blocker of the xenobiotic efflux pumps encoded by *Bcrp1* and *Mdr1a/b*)[36] or cell death (treatment with KN-62, which prevents cell death triggered by ATP and/or NAD-induced activation of the P2RX7 receptor during tissue

**Fig. 1 | CD8+ T cells in the gut LP and epithelial layer have remodeled the transcription of mitochondria-encoded genes. a** Schematic representation of the gut-specific CD8+ T cell response and strategy designed to investigate it. Image created with Biorender.com. **b** Uniform Manifold Approximation and Projection (UMAP) of the single cell RNA and ADT sequencing data obtained from the experiments designed as in (**a**). Upon clustering analysis, 10 RNA-based clusters (0–9) were identified, using a resolution of 0.2. 49,111 cells obtained from 4 tissues and 3 biological replicates were analyzed. Color-coding is indicated in the legend. **c** UMAP of the single cell RNA and ADT sequencing data. Color-coding is according to the tissue of origin as outlined in the legend. The labeling in quotation marks refers to the data in (**b**). **d** Bar graph of the distribution of the clusters identified in (**b**) across different tissues of origin. Within every tissue, cell numbers are

normalized to 100% and the fractional contribution of every cluster is indicated. Color-coding resembles the one used in (**b**). **e** UMAP of the single cell RNA and ADT sequencing data showing the ratio of mitochondrial transcripts over the total transcriptome, across different clusters. The labeling in quotation marks refers to the data in (**b**). Violin plots show the fraction of mitochondria-derived transcripts over the total transcriptome, across different clusters. The black dashed line indicates the median value identified for cluster 6. **f** Plots representing the expression of selected genes as a function of pseudotime, as identified in the data shown in Supplementary Fig. 2a. On the *x* axis, the interquartile range of every cluster is indicated, to facilitate the correlation between gene expression and progression of cells through pseudotime. Color-coding is indicated in the legend.

---

preparation) between mLN and CD8+ T cells isolated from the two intestinal compartments[37] (Supplementary Fig. 3d, e). Adding to the Mitotracker green data, western blot analysis showed reduction of the mitochondrial structural protein TOM20 and slightly reduced expression of the mitochondria-specific transcription factor TFAM in CD8+ T cells isolated from the LP and IEL as compared to cells from the mLN (Fig. 2c and Supplementary Fig. 3f).

To confirm our results in an antigen-specific setting, we transferred CD90.1+ ovalbumin (OVA)-specific CD8+ T cells into CD90.1− recipient mice, and infected them orally with a transgenic strain of *Listeria monocytogenes* expressing OVA (*LmOVA*), to study the OVA-specific intestinal CD8+ T cell response. OVA-specific CD8+ T cells isolated 7 and 24 days post-infection from the LP and IEL of infected mice showed substantially reduced mitochondrial content as compared to OVA-specific cells isolated from the mLN, mirroring the results obtained from the analysis of the polyclonal response (Fig. 2d and Supplementary Fig 4a).

Since mitochondrial dynamics (fission vs fusion) have been linked to T cell fate acquisition[4], we isolated CD8+ T cells from the mLN, LP, and IEL of PhAM mice, expressing the mitochondria-localized version of the fluorescent protein Dendra2[38], and assessed their mitochondrial morphology using live cell imaging. We found that CD69+CD103+ cells isolated from the LP and IEL showed reduced numbers of mitochondria as compared to spleen- and mLN-isolated T cell populations (Supplementary Fig. 4b). Moreover, mitochondria of CD8+ T cells isolated from the LP and the IEL fraction were fragmented and shared similarities with mLN-isolated $T_{EM}$, as compared to $T_N$ or $T_{CM}$, previously shown to form complex networks of fused mitochondria[4] (Supplementary Fig. 4b). Western blot analysis of the distribution of OPA-1 isoforms, with the relative accumulation of the shortest being associated with mitochondrial fragmentation[39], correlated with the data obtained from PhAM mice (Supplementary Fig. 4c). Finally, transmission electron microscopy analysis to assess mitochondrial ultrastructure showed that mitochondria present in LP and epithelia-isolated cells had dilated cristae, as compared to mLN-resident cells (red arrowheads, Supplementary Fig. 4d), supporting the data regarding the distribution of the OPA-1 isoforms. Altogether, these data corroborate the finding that CD8+ T cells isolated from the LP and epithelial layer of the gut have a fragmented mitochondrial network, as compared to CD8+ T cells localized in the mLN.

We next tested whether reduced mitochondrial mass was accompanied by defective mitochondrial function. We used TMRM to assess mitochondrial membrane potential across the inner mitochondrial membrane, as a readout of the functionality of the electron transport chain (ETC) and movement of protons across the inner mitochondrial membrane. Analysis of CD8+ T cells across the physiological intestinal immune response showed that LP- and IEL-isolated CD8+ T cells had reduced mitochondrial membrane potential when compared to cell populations in the mLN, mirroring the staining pattern observed for Mitotracker green (Fig. 2e). Similarly, we found that the mitochondrial membrane potential of antigen-specific CD8+ T cell subsets in the LP and epithelial layer of the gut was reduced in comparison to the one

recorded in mLN-isolated cells (Fig. 2f). Notwithstanding the reduction of mitochondrial mass, LP CD8+ T cells were able to effectively engage the ETC in the remaining mitochondria to produce ATP, as shown by TMRM staining increasing after inhibition of the ATP synthase by oligomycin, an effect dissipated by the uncoupling agent FCCP (Fig. 2g). CD69+CD103+ T cells isolated from the gut LP, as well as epithelial layer, had a significantly lower ATP/AMP ratio, but did not have significant defects in their energy balance, as measured by AMP, ADP, and ATP levels, as compared to cells isolated from mLN (Fig. 2h). To assess the potential impact of mitochondrial loss over the function of CD8+ T cells, we analyzed a mouse model that uses eYFP to report one-to-one interferon-γ protein expression, without requiring restimulation of cells (GREAT mice[40]). We found that the reduction of mitochondrial mass observed in LP- and IEL-isolated CD8+ T cells did not necessarily lead to defective cytokine production, as most of the cells isolated from the LP and IEL produced interferon-γ (Fig. 2i).

To complement our analysis, we performed in-depth transcriptional profiling using bulk RNA sequencing of the different CD8+ T cell populations. Principal component analysis (PCA) of their transcriptome positioned the cell populations in a progression from more quiescent populations (mLN-isolated $T_N$, $T_{CM}$, and $T_{RM}$) to activated ones (mLN-isolated $T_{EM}$) and ultimately to populations isolated from the intestinal wall (LP and IEL-isolated CD69−CD103− cells and CD69+CD103+ cells) (Supplementary Fig. 4e). As with our single cell analyses, we observed that $T_N$ and $T_{CM}$ clustered together, and were separated from the tissue populations by $T_{EM}$. Moreover, LP populations laid between mLN and IEL cells, thus validating our earlier observations and trajectories (Supplementary Fig. 4e). When we analyzed the differentially expressed genes (DEGs) across the cell subsets, we found that adaptation of CD8+ T cells to the LP triggered a transient global upregulation of transcripts (transition 3), whereas acquisition of the CD69+CD103+ phenotype (transitions 4 and 5) caused downregulation of the transcriptome (Supplementary Fig. 4f). Moreover, bulk RNA sequencing confirmed the transcriptional signature identified in Fig. 2, as transcripts involved in mitochondrial function showed broad downregulation during the transition between mLN $T_{EM}$ and LP CD69+CD103+ cells (Supplementary Fig. 4g). Overall, we found that adaptation of CD8+ T cells to the intestinal microenvironment correlates with reduced mitochondrial mass, membrane potential, and fragmentation of the mitochondrial pool. In spite of this, CD8+ T cells isolated from both the LP and epithelial layer of the gut maintained functionality of the remaining mitochondria to sustain a viable energy balance and cytokine production.

## Prostaglandin E₂ regulates mitochondrial content and fitness of intestinal LP and epithelial T cells

CD8+ T cells are distributed across nearly all peripheral organs. To test if the reduction of mitochondrial content is a phenomenon restricted to the gut or occurs in other tissues, we measured Mitotracker green uptake in CD8+ T cells isolated from the parenchyma of liver and lung. We found that gut LP-isolated CD69+CD103+ cells uniquely displayed substantially reduced mitochondrial mass as compared to

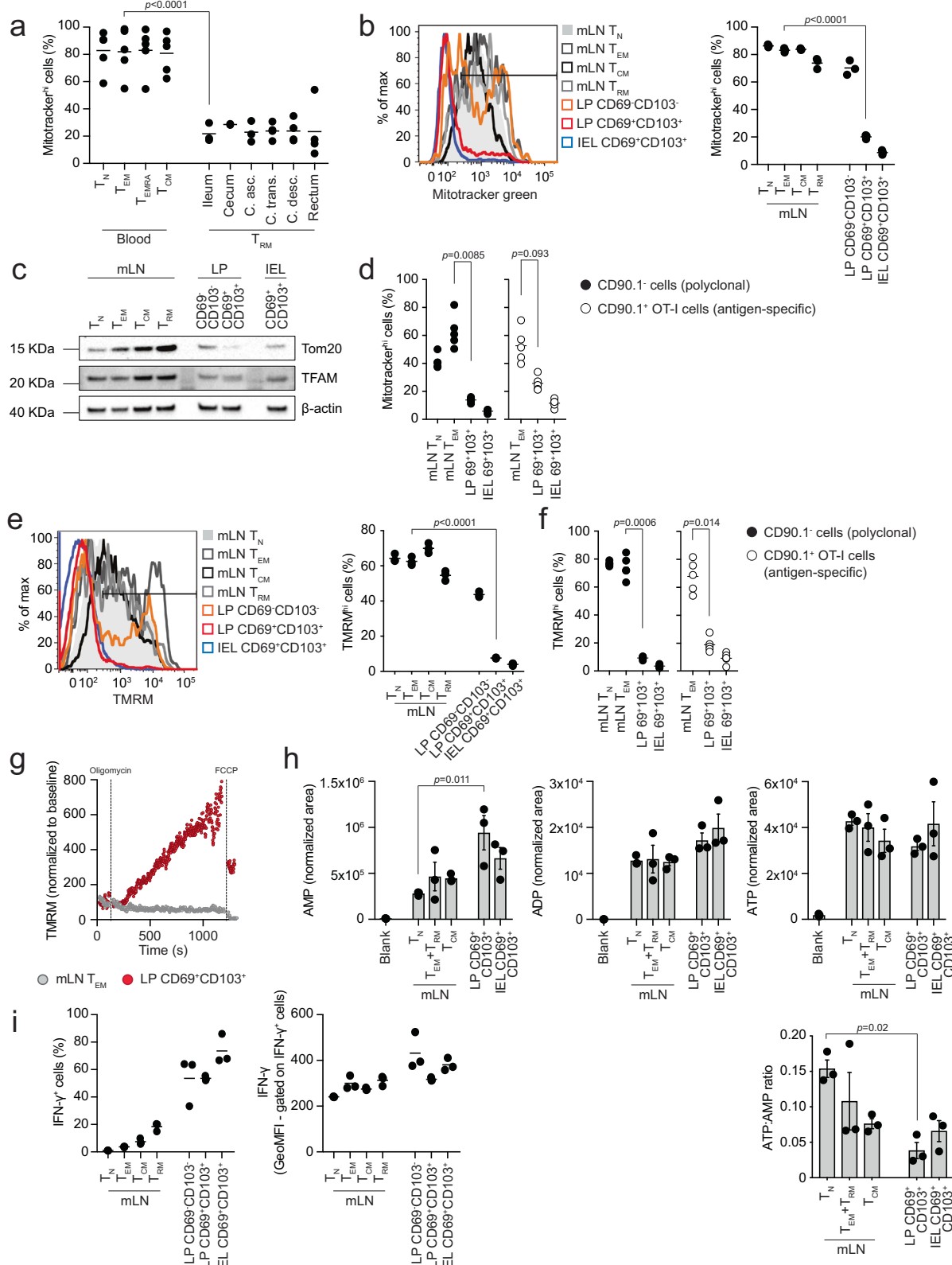

counterparts isolated from mLN as well as from liver and lung (Fig. 3a). Interestingly, gut LP-resident macrophages did not show any reduction of mitochondrial content as compared to other tissues, suggesting that the phenotype was confined to CD8[+] T cells isolated from the intestinal compartments (Supplementary Fig. 5a).

To investigate how the intestinal microenvironment might regulate the mitochondrial content of CD8[+] T cells, we isolated the interstitial fluid of several tissues and analyzed the fluid using untargeted metabolomics. The analysis revealed a group of metabolites uniquely present in the small intestine, including prostaglandin $E_2$ ($PGE_2$) (Fig. 3b, c). $PGE_2$ signaling is constitutive in the gut and it contributes to tissue repair and maintenance of the epithelial barrier[41,42]. Also, we recently reported that $PGE_2$ controls mitochondrial membrane potential in macrophages[43]. To test the role of $PGE_2$ in driving

**Fig. 2 | Intestinal LP and IEL T cells have reduced mitochondrial content. a** Flow cytometry analysis of Mitotracker green in cell populations from blood and intestine of healthy humans. Lines in the dot plot show mean values and graph shows data from 2 to 5 independent experiments. **b** Mitotracker green in cell populations isolated from mLN and intestine of C57BL/6J mice. Lines in the dot plot show mean values and data show $n = 3$ biological replicates over nine independent experiments. **c** Western blot of Tom20 and TFAM. The data shown are representative of six independent experiments. **d** Mitotracker green in antigen-specific and polyclonal cell populations isolated from mLN and intestine of mice orally challenged with *LmOVA* and analyzed 7 days post-infection. Lines in the dot plot show mean values and data show $n = 5$ biological replicates over four independent experiments. **e** TMRM staining in the indicated cell populations. Lines in the dot plot show mean values and data show $n = 3$ biological replicates over five independent experiments. **f** TMRM in antigen-specific and polyclonal cells isolated from mLN

and intestine of mice orally challenged with *LmOVA* and analyzed 7 days post-infection. Lines in the dot plot show mean values and data show $n = 5$ biological replicates over four independent experiments. **g** TMRM staining in cells upon treatment with oligomycin and FCCP. TMRM was used at a concentration of 10 nM. TMRM values are shown normalized to the baseline of staining in the respective population. Data are representative of three independent experiments. **h** Mass spectrometry analysis of AMP, ADP, ATP, and ATP:AMP ratio. The bars show mean ± SEM. Data show three independent experiments analyzed simultaneously. **i** Intracellular interferon-γ (IFN-γ) protein expression in cells isolated from mLN and intestine of GREAT mice. Lines in the dot plots show mean values and data show $n = 3$ biological replicates over two independent experiments. **a, b, d–f,** and **h** Statistics were performed using one-way ANOVA and Tukey's multiple comparison correction. Source data are provided as a Source Data file.

the reduction of mitochondrial content in CD8$^+$ T cells, we established an in vitro model to partly mimic the exposure of CD8$^+$ T cells to the tissue environment, and we named these cells (IL-15/TGF-β)-T cells (Supplementary Fig. 5b). Treatment with IL-15 and TGF-β induced a phenotype characterized by high expression of the integrin CD103, as compared to cells treated with IL-2 [(IL-2)-T cells] (Supplementary Fig. 5c). (IL-15/TGF-β)-T cells substantially reduced their mitochondrial mass and membrane potential upon PGE$_2$ exposure, similar to what we observed for CD8$^+$ T cells isolated from the LP and epithelial layer of the intestine (Fig. 3d). Also, (IL-15/TGF-β)-T cells seemed to be more sensitive to the effect of PGE$_2$ treatment on mitochondrial content, as compared to (IL-2)-T cells (Supplementary Fig. 5d, e). PGE$_2$ was able to drive reduction of the mitochondrial content upon IL-15/TGF-β and IL-15-alone conditions (Supplementary Fig. 5f), suggesting that TGF-β signaling, at least in our setting, is not required to mediate the effect of PGE$_2$ on the metabolic fitness of CD8$^+$ T cells. The effect of PGE$_2$ was dependent on its concentration, with as little as 10 nM PGE$_2$ triggering the reduction of mitochondrial mass and membrane potential in (IL-15/TGF-β)-T cells (Supplementary Fig. 5g). Finally, the reduction of mitochondrial content in (IL-15/TGF-β)-T cells was specifically triggered by PGE$_2$ and PGE$_1$, while exposure to other prostaglandins did not cause any noticeable effect (Fig. 3e). Since PGE$_2$ and PGE$_1$ both signal through EP1-4 receptors, we analyzed receptor expression in vivo across CD8$^+$ T cell subsets. We found that *Ptger4*, encoding for the EP4 receptor, was uniquely upregulated in CD8$^+$ T cells isolated from the LP and IEL fraction (Supplementary Fig. 5h). Moreover, the single cell data confirmed that *Ptger4* transcripts were substantially upregulated in cells occupying the UMAP space of LP and IEL, as compared to cells positioned in the "mLN area" (Fig. 3f). CRISPR-Cas9 mediated *Ptger4* deletion impinged on the ability of (IL-15/TGF-β)-T cells to sense PGE$_2$ and partially rescued the reduction of mitochondrial content and function (Supplementary Fig. 5i, j).

To investigate whether PGE$_2$-driven reduction of mitochondrial content controls the adaptation of CD8$^+$ T cells to the gut microenvironment and the establishment of the T cell pool, we tested the effect of *Ptger4* deletion in the generation of intestinal CD8$^+$ T cells following oral *Listeria monocytogenes* infection. Recipient mice were co-transferred with control (Ctrl) or *Ptger4*-deficient naive ovalbumin (OVA)-specific CD8$^+$ T cells that expanded in response to oral challenge with a transgenic strain of *Listeria monocytogenes* expressing ovalbumin (*LmOVA*) (Fig. 3g). *Ptger4*-deficient cells outcompeted control cells in establishing the antigen-specific pool of CD8$^+$ T cells in the LP and in the induced-IEL fraction, upon oral infection with *LmOVA*, whereas no such difference was observed in mLN (Fig. 3h, i, Supplementary Fig. 5k). Moreover, *Ptger4*-deficient CD69$^+$CD103$^+$ cells isolated from LP showed higher mitochondrial content, as compared to control counterparts (Fig. 3j). Interestingly, *Ptger4*-deficient cells isolated from the LP were more prone to retain the CD69$^-$CD103$^-$ phenotype, as compared to control cells, and fewer of them acquired expression of CD69 (Fig. 3k). On the other hand, deletion of *Ptger4* only minimally

affected the ability of LP and epithelium-isolated CD8$^+$ T cells to secrete interferon-γ and TNF (Supplementary Fig. 5l). These data show that the loss of mitochondrial content is, at least in part, a consequence of being in an environment rich in PGE$_2$, and disruption of PGE$_2$ sensing promotes the accumulation of CD8$^+$ T cells in the intestinal LP and epithelial layer. Altogether, we found that the intestinal microenvironment uniquely triggered the reduction of the mitochondrial content of CD8$^+$ T cells. We also identified PGE$_2$ as one of the causative agents initiating this process.

## Autophagy and glutathione counterbalance the effect of PGE$_2$ and maintain the pool of T cells in the LP and epithelial layer of the gut

Next, we addressed the issue of how PGE$_2$ exposure influenced the mitochondrial content of CD8$^+$ T cells. Mitochondria are key to regulate cell function. These organelles routinely undergo quality control to test their fitness, and dysfunctional mitochondria are cleared by mitophagy, i.e., mitochondrial autophagy (reviewed in ref. 44). A drop in mitochondrial membrane potential has been shown to trigger clearance of depolarized mitochondria[45]. Since we showed that PGE$_2$ contributed to the reduction of mitochondrial membrane potential, we hypothesized that mitochondrial depolarization may trigger autophagy-mediated clearance of mitochondria in CD8$^+$ T cells exposed to the gut environment. Indeed, CD8$^+$ T cells upregulated most of the autophagy-related genes (*Atg* genes) as well as *Optn*, *Prkn*, *Sqstm1*, *Ulk1*, and *Pink1*, associated with mitophagy, during their transition from mLN to the LP and IEL compartments of the intestine (Fig. 4a). CD69$^+$CD103$^+$ cells isolated from LP and IEL expressed the lipidated form of LC3 (LC3-II), uniquely formed upon autophagy initiation (Fig. 4b). Treating LP- and IEL-isolated CD8$^+$ T cells with the autophagy inhibitor bafilomycin A1 inhibited autophagic flux and drove accumulation of the lipidated form of LC3, suggesting that LP- and IEL-isolated CD69$^+$CD103$^+$ cells actively engaged autophagy (Supplementary Fig. 6a). Moreover, CD8$^+$ T cells isolated from LP and IEL showed reduced expression p62, encoded by *Sqstm1*, which is degraded in the autophagy process (Fig. 4b and Supplementary Fig. 6b). Finally, transmission electron microscopy provided further evidence that LP and IEL-isolated CD8$^+$ T cells upregulated autophagy, as structures resembling autophagosomes were identified only in images of LP CD69$^+$CD103$^+$ and IEL CD69$^+$CD103$^+$ cells (Fig. 4c). Treating (IL-15/TGF-β)-T cells with bafilomycin A1 partially blunted the reduction of mitochondrial mass observed upon PGE$_2$ treatment, but failed to rescue the reduction of mitochondrial membrane potential, suggesting that inhibition of autophagy prevented mitochondrial clearance while leaving PGE$_2$ sensing and the subsequent drop of mitochondrial membrane potential unaffected (Fig. 4d). Inhibition of autophagy with wortmannin, a PI3K inhibitor, or SBI-0206965, an ULK1 inhibitor, equally prevented mitochondrial clearance upon treatment with PGE$_2$ (Supplementary Fig. 6c, d). However, both wortmannin and SBI-0206965

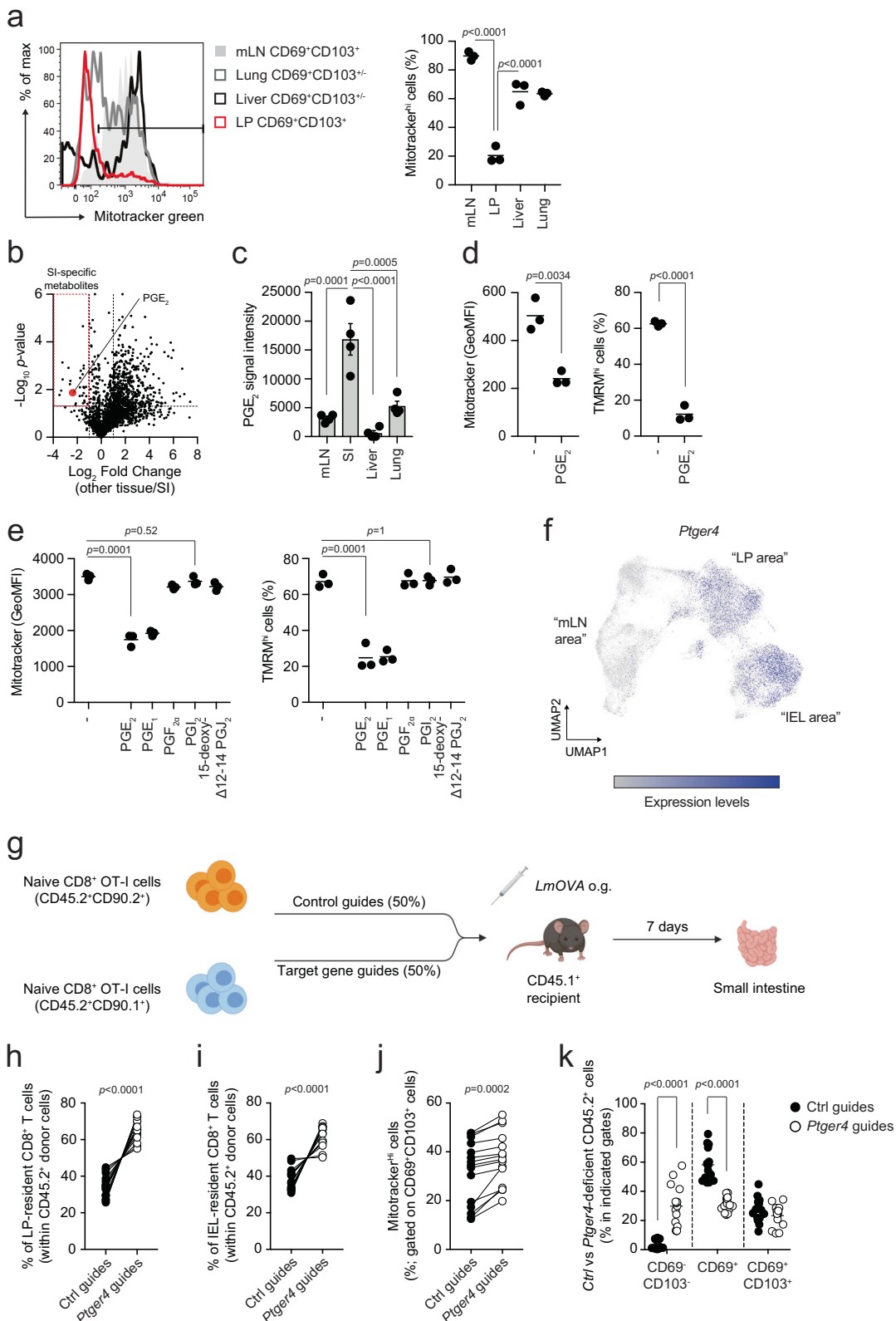

also rescued the PGE$_2$-driven reduction of mitochondrial membrane potential (Supplementary Fig. 6c, d). Finally, we tested whether autophagy plays a role in the formation of the pool of gut CD8$^+$ T cells upon oral challenge with *LmOVA*. We blocked autophagy by deleting *Atg5* with CRISPR-Cas9 (Supplementary Fig. 6e) and found that *Atg5*-deficient CD8$^+$ T cells failed to establish the antigen-specific pool of CD8$^+$ T cells in LP and induced IEL fraction, as compared to control

cells (Fig. 4e, f). No consistent changes in the distribution of Ctrl vs *Atg5*-deficient cells were observed in the pool of antigen-specific CD8$^+$ T cells in mLN (Supplementary Fig. 6f). In line with our in vitro results (Fig. 4d), blocking autophagy partially prevented mitochondrial clearance in *Atg5*-deficient LP CD69$^+$CD103$^+$ cells, as compared to controls (Fig. 4g). These results suggest that failure to clear depolarized mitochondria in intestinal, PGE$_2$-exposed CD8$^+$ T cells

**Fig. 3 | Prostaglandin E₂ regulates mitochondrial content of intestinal LP and IEL-isolated CD8⁺ T cells. a** Mitotracker green in CD8⁺CD69⁺CD103⁺/⁻ cells isolated from the tissues of unchallenged mice. Lines in the dot plot show mean values and the data show $n = 3$ biological replicates over two independent experiments. **b** Volcano plot shows the differential distribution of metabolites isolated from the interstitial fluid of tissues. Dashed lines indicate the fold change filter of FC > 2 and FC < −2 and the $p$ value filter of $p = 0.05$. Plot shows cumulative data of four independent experiments. Statistics were performed using Student's $t$ test between metabolites identified in the small intestine and metabolites identified in all the other tissues. No FDR correction was applied. **c** Relative abundance of PGE₂ in interstitial fluid isolated from the indicated tissues. Graph shows mean values ± SEM of cumulative data of four independent experiments. **d** Mitotracker green and TMRM in CD8⁺ T cells activated in (IL-15/TGF-β)-T cell-polarizing conditions for 5 days and treated for 24 h with 10 μM PGE₂. Lines in the dot plot show mean values and the data show $n = 3$ biological replicates over three independent experiments. Statistics were performed using two-tailed Student's $t$ test. **e** Mitotracker green and TMRM staining in CD8⁺ T cells activated in (IL-15/TGF-β)-T cell-polarizing conditions for 5 days and treated for 24 h with 100 nM of different prostaglandins. Lines in the dot plot show mean values and the data show $n = 3$ biological replicates over two independent experiments. Statistics were performed using one-way ANOVA and Dunnet's multiple comparison correction. **f** UMAP of the single cell RNA and ADT sequencing data showing expression across different clusters of *Ptger4*. **g** Schematic of the *Listeria monocytogenes* expressing OVA (*LmOVA*) infection model used in Figs. 3–5. OT-I donor cells were CD45.2⁺, whereas recipient mice were CD45.1⁺. CD90.1 and CD90.2 were used to additionally separate control guide- *vs* target gene guides-treated donor cells. *LmOVA* was administered by oral gavage. Image created with Biorender.com. **h**–**i** Distribution of Ctrl vs *Ptger4*-deleted CD8⁺ T cells within the population of CD45.2⁺ cells upon *LmOVA* challenge in LP (**h**) and IEL fraction (**i**). Lines in the dot plot show pairing within single mice, and the dot plots show cumulative data of $n = 15$ biological replicates over three independent experiments. **j** Mitotracker green in Ctrl vs *Ptger4*-deleted CD69⁺CD103⁺ cells isolated from LP of mice orally challenged with *LmOVA*. Lines in the dot plot show pairing within single mice, and dot plots show cumulative data of $n = 15$ biological replicates over three independent experiments. **k** Distribution of Ctrl vs *Ptger4*-deleted CD8⁺ T cells gated in the population of CD45.2⁺ cells between CD69⁻CD103⁻, CD69⁺ and CD69⁺CD103⁺ populations, upon *LmOVA* challenge in LP. Lines in the dot plot show mean values; dot plots show cumulative data of $n = 15$ biological replicates over three independent experiments. Statistics were performed using two-way ANOVA and Sidak's multiple comparison correction. **a**, **c** Statistics were performed using one-way ANOVA and Tukey's multiple comparison correction. **h**, **i** Statistics were performed using two-tailed paired $t$ test. Source data are provided as a Source Data file.

---

negatively impacts the fitness of the intestinal CD8⁺ T cell pool in the LP and epithelial layer.

We wondered whether the PGE₂-driven reduction in mitochondrial membrane potential resulted in the production of reactive oxygen species (ROS), and thus altered redox balance, in intestinal CD8⁺ T cells. We analyzed Mito-roGFP2-Orp1 mice, encoding a mitochondria-targeting redox-sensitive GFP that senses hydrogen peroxide (H₂O₂)[46,47] and found that CD8⁺ T cells isolated from the intestinal LP and IEL fraction exhibited an increased proportion of cells with oxidized mitochondria, as compared to $T_N$ and $T_{CM}$ cells isolated from mLN, while $T_{EM}$ had the most oxidized mitochondria. (Fig. 4h and Supplementary Fig. 6g). Although LP and IEL T cells had increased mitochondrial ROS, as quantified in Mito-RoGFP2-Orp1 mice, these cells had low cellular ROS, as quantified by CellROX, compared to the CD8⁺ T cells resident in the mLN (Fig. 4i). Reconciling these seemingly contradicting data, we found that CD8⁺ T cells isolated from LP and IEL upregulated a panel of genes associated with "antioxidant activity", including genes encoding for glutathione peroxidases (*Gpx* genes), transferases (*Gstm* genes) and reductase (*Gsr*) (Fig. 4j). Both the reduced (GSH) and oxidized (GSSG) forms of glutathione were elevated in CD69⁺CD103⁺ T cells isolated from LP and IEL compared to T cells isolated from the mLN, while the GSH/GSSG ratio was unaltered (Fig. 4k and Supplementary Fig. 6h). These results suggested that upon sensing the gut microenvironment, CD8⁺ T cells enhance their antioxidant defenses to cope with elevated mitochondrial ROS and maintain a viable intracellular redox balance. To assess the contribution of glutathione in the formation of the intestinal CD8⁺ T cell pool in gut LP and epithelial layer, we genetically ablated the catalytic subunit of glutamate cysteine ligase (*Gclc*), an enzyme critical for the de novo synthesis of glutathione, which has previously been shown to be key to T cell development and function[48,49] (Supplementary Fig. 6i). Similar to what we observed upon autophagy inhibition, *Gclc* deletion remarkably affected accumulation of CD8⁺ T cells in LP and induced-IEL fractions in mice orally challenged with *LmOVA* (Fig. 4l, m). Glutathione thus appears to be key to sustain the redox balance and viability of the LP and IEL CD8⁺ T cell pool and to counterbalance the accumulation of ROS resulting from PGE₂-driven mitochondrial depolarization.

## Got1 links PGE₂ with the regulation of mitochondrial content in intestinal LP and epithelial CD8⁺ T cells

To investigate the mechanisms underlying the regulation of mitochondrial content mediated by PGE₂, we first assessed whether PGE₂ itself was sufficient to promote autophagy. Treatment of (IL-15/TGF-β)-T cells with 100 nM PGE₂ induced accumulation of LC3-II, as compared to untreated cells (Fig. 5a), suggesting that PGE₂ induces autophagy in CD8⁺ T cells.

As CD8⁺ T cells in the LP and IEL fraction of the intestine had a fragmented mitochondrial network (Supplementary Fig. 4), and mitochondrial fission plays a role in mitophagy, we tested whether *Drp1* was required to mediate the PGE₂-driven reduction of mitochondrial content[50]. Mice carrying a *Drp1* deletion in the T cell compartment (*Drp1*^f/f^*CD4*^Cre+^) showed a normal distribution of CD8⁺ T cells across mLN, LP and IEL compared to littermate controls (*Drp1*^f/f^*CD4*^Cre^) (Fig. 5b). Furthermore, deficiency of *Drp1* did not affect the ability of LP and IEL-isolated CD8⁺ T cells to reduce their mitochondrial content and function in the gut microenvironment (Fig. 5c, d). These data suggest that *Drp1*-deficient cells are able to respond normally to PGE₂ in the gut, and are in line with previous reports suggesting that *Drp1*-mediated mitochondrial fission could be dispensable for mitophagy[51–53].

We used hints from our previous work[43] as a framework to further explore PGE₂ regulation of mitochondrial content in intestinal LP and epithelial CD8⁺ T cells. Bulk RNA sequencing showed a strong upregulation of *Got1* in CD69⁺CD103⁺ cells isolated from the LP (Fig. 5e). This result was confirmed at the single cell level (Fig. 5f) and, of note, both the datasets highlighted that *Got1* upregulation occurred only in the LP, sparing IEL (Fig. 5e, f). *Got1* encodes an enzyme playing a key role in the activity of the malate-aspartate shuttle (MAS), a system that controls redox balance between mitochondria and cytosol. While other enzymes involved in the MAS were not substantially affected in this setting, the MAS metabolite NAD⁺ was decreased in CD8⁺ T cells isolated from the LP and IEL fraction of the gut compared to mLN-isolated counterparts (Fig. 5g). Reduced NAD⁺ levels likely affected function of the NAD⁺-dependent enzyme GAPDH, key to regulate glycolytic flux. LP and IEL-isolated CD8⁺ T cells accumulated metabolites upstream of GAPDH, such as hexose phosphate, fructose 1,6-bisphosphate and dihydroxyacetone phosphate (DHAP), while the levels of phosphoenolpyruvate (PEP), downstream of GAPDH, remained unaffected (Supplementary Fig. 7a). All in all, these data build upon our previous work[43] and implicate that intestinal PGE₂ may affect the function of MAS, and ultimately shape mitochondrial content and function in CD8⁺ T cells. To test this hypothesis, we deleted *Got1* in OVA-specific cells (Supplementary Fig. 7b), transferred them in recipient mice and assessed if *Got1* played a role in controlling mitochondrial content and function in CD8⁺ T cells isolated from the LP and IEL, upon challenge with *LmOVA*. While *Got1* deficiency did not consistently affect the distribution of CD8⁺ T cells in the LP and IEL (Supplementary Fig. 7c), it did affect the ability of CD8⁺ T cells to

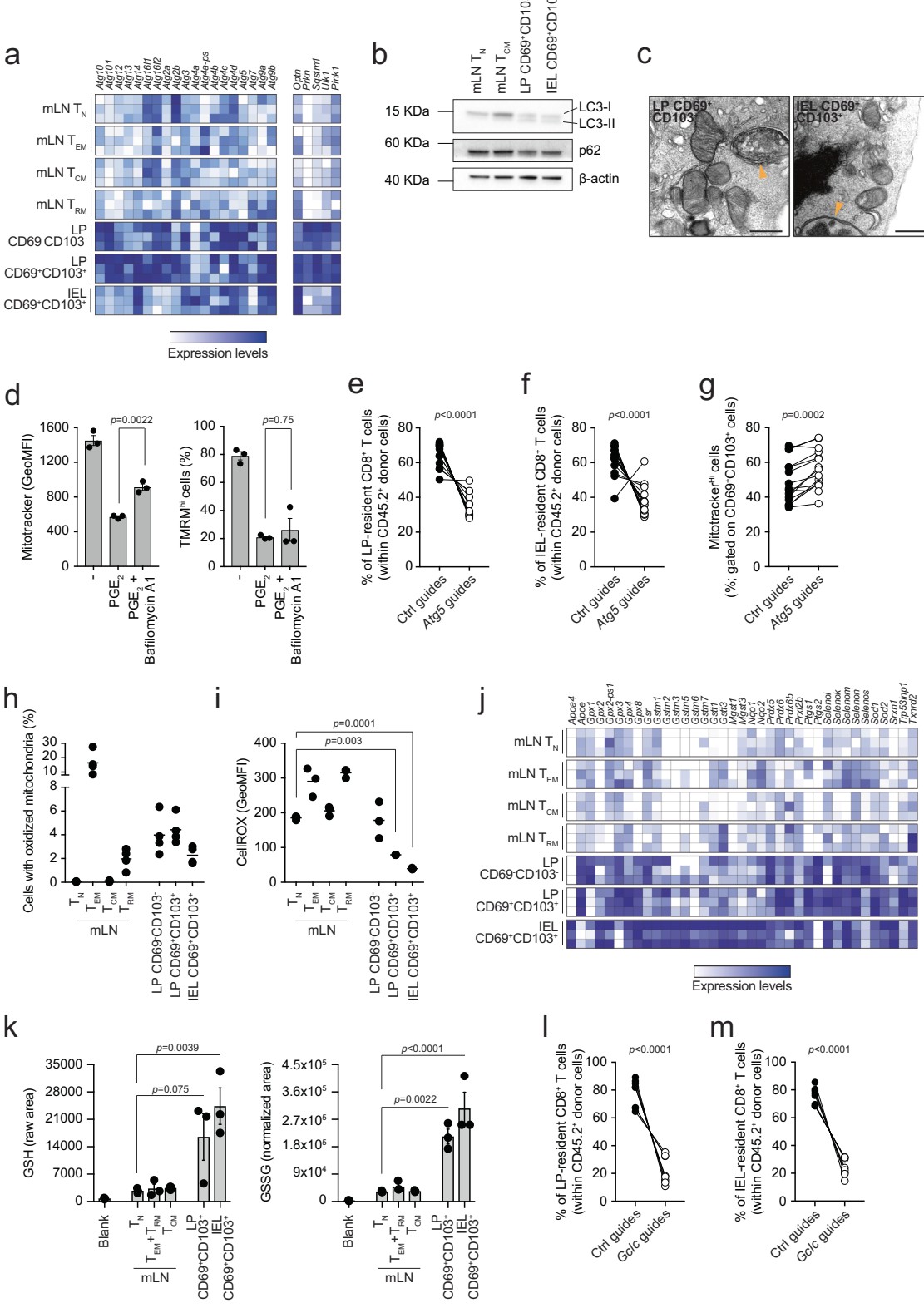

reduce their mitochondrial content and function in response to gut LP entry (Fig. 5h, i). The impairment of the reduction of mitochondrial content was observed only in CD69−CD103− and CD69+ T cells, suggesting that additional mechanisms other than the one mediated by *Got1* could regulate mitochondrial content in CD8+ T cells. Of note, *Got1*-deficient cells were less prone to transit to the CD69+ state, and accumulated as CD69−CD103− cells (Fig. 5j), reminiscent of the effect driven by *Ptger4* deletion (Fig. 3k).

In summary, we identified a mechanism that underlies the adaptation of CD8+ T cells to the intestinal microenvironment. T cells entering the LP and epithelial layer of the gut sense PGE2, expressed in the gut during homeostasis, which drives reduction of mitochondrial mass and membrane potential in these CD8+ T cells, which express EP4. We propose that PGE2 affects the mitochondrial membrane potential via alterations of the malate-aspartate shuttle, autophagy contributes to the clearance of depolarized mitochondria, whereas glutathione

**Fig. 4 | Autophagy and glutathione maintain the fitness of LP and IEL T cells.**
**a, j** Heatmaps show the expression of selected genes of bulk RNA sequencing data. Color-coding is as per legend (white, low expression; blue, high expression) and shows relative values within each gene. Data show three independent experiments. **b** Western blot of LC3 isoforms and p62. The data are representative of two independent experiments (LC3) or one experiment (p62). **c** Transmission electron microscopy of cells isolated from unchallenged mice. Yellow arrowheads indicate double membrane structures characteristic of autophagosomes. Scale bar = 500 nm. The panel shows representative images of one experiment. **d** Mitotracker green and TMRM in CD8$^+$ T cells activated in (IL-15/TGF-β)-T cells polarizing conditions for 5 days and treated with PGE$_2$ in the presence of bafilomycin A1. Lines in the dot plot show mean values ± SEM and the data show $n = 3$ biological replicates over five independent experiments. **e−f** Distribution of Ctrl vs *Atg5*-deleted CD8$^+$ T cells within the population of CD45.2$^+$ cells upon *LmOVA* challenge in LP (**e**) and IEL fraction (**f**). Lines in the dot plot show pairing within single mice, and the dot plots show cumulative data of $n = 15$ biological replicates over three independent experiments. **g** Mitotracker green in Ctrl vs *Atg5*-deleted CD69$^+$CD103$^+$ cells

isolated from LP of mice orally challenged with *LmOVA*. Lines in the dot plot show pairing within single mice, and dot plots show cumulative data of $n = 15$ biological replicates over three independent experiments. **h** Mitochondrial oxidative state, as indicated by a shift from green to blue fluorescence, in cells isolated from mLN and intestine of Mito-roGFP2-Orp1. Lines in the dot plot show mean values and the data show n = 4 biological replicates over two independent experiments. **i** CellROX staining. Lines in the dot plot show mean values and the data show $n = 3$ biological replicates over three independent experiments. **k** Mass spectrometry analysis of levels of GSH and GSSG in the indicated cell populations. The bars show mean ± SEM. Data show three independent experiments analyzed simultaneously.
**l−m** Distribution of Ctrl vs *Gclc*-deleted CD8$^+$ T cells within the population of CD45.2$^+$ cells upon *LmOVA* challenge in LP (**l**) and IEL fraction (**m**). Lines in the dot plot show pairing within single mice, and the data show $n = 8$ biological replicates over two independent experiments. **d, h, i** and **k** Statistics were performed using one-way ANOVA and Tukey's multiple comparison correction. **e−g, l, m** Statistics were performed using two-tailed paired $t$ test. Source data are provided as a Source Data file.

maintains the redox balance in PGE$_2$-responding cells to ultimately shape the pool of gut CD8$^+$ T cells (Fig. 5k).

## Discussion

Immune responses are initiated in lymphoid tissues, but immune cells carry out their effector function in peripheral tissues. Thus, understanding the adaptation of immune cells to their surrounding environment is key to design new approaches to regulate immunity in a tissue-specific fashion. Investigation of tissue immunity has been hindered by technical limitations that constrain the yield of cells that can be isolated from these tissues and analyzed[54]. Single cell technologies are now providing a way to partially overcome these limitations, and to capture transcriptional changes that may underscore the adaptation of cells to the tissue microenvironment. In this study, we used single cell RNA sequencing and oligo-tagged antibody staining to characterize how the intestine-specific CD8$^+$ T cell response unfolds across lymphoid (mLN) and peripheral (LP and IEL) tissues. Of note, our analysis focuses on thymus-selected CD8αβ T cells, from their naïve stage in mLN, to their terminally-differentiated stage as induced IEL. We took advantage of the constitutive physiological activation of the gut immune response driven by microbiota and food antigens (reviewed in ref. 55) to investigate the adaptation of T cells to the gut microenvironment. By using this strategy, we overcame the potential alterations of tissue architecture caused by an overt infection that could potentially affect the homeostatic environment of the gut. However, to confirm our findings in an antigen-specific model, we orally challenged mice with *L. monocytogenes*, to assess the adaptation of antigen-specific CD8$^+$ T cells to the gut, and to test how genetic manipulations of the PGE$_2$-autophagy-glutathione axis affected accumulation of CD8$^+$ T cells in the intestine.

Single cell analysis of the gut CD8$^+$ T cell immune response identified a continuum of transcriptional and surface features that positioned the lamina propria (LP) of the small intestine as a transit site for recently-activated CD8$^+$ T cells (T$_{EM}$) trafficking from the mLN to the periphery. Despite the limitations inherent to this analysis, such as the "snapshot" nature of it, this finding is in line with the notion that blood vessels that supply the small intestine are embedded within the LP[56], suggesting that this tissue acts as a seeding ground for the whole gut, and is the location where T cell adaption to the gut environment likely occurs. Our analysis found that gut adaptation is underscored by transcriptional and metabolic changes, prompting the questions of whether and how the gut microenvironment affects the epigenetic landscape of CD8$^+$ T cells exposed to it. Metabolites such as acetate, produced by the microbiota, may affect the intracellular pool of acetyl coenzyme A that is used as a substrate for acetylating histones and thus regulating gene expression. This may also be an interesting topic of study in light of

the ability of gut-resident T$_{RM}$ to rejoin the systemic circulation and immune responses[57].

The major metabolic change we identified upon adaptation of CD8$^+$ T cells to the gut environment was a stark reduction in mitochondrial content driven by sensing of prostaglandin E$_2$ (PGE$_2$). We suggest that T cells clear mitochondria following a drop in mitochondrial membrane potential driven by PGE$_2$ and its regulation of the malate-aspartate shuttle. Despite mitochondrial depolarization and clearance, intestinal LP and epithelial T cells showed a viable energy balance and the remaining mitochondria could efficiently engage the electron transport chain. Our analysis did not show whether the entire mitochondrial repertoire of a cell was homogeneously affected by PGE$_2$. It is likely that depolarized *vs* polarized mitochondria are sorted for clearance through autophagy, and it would be interesting to study if and how mitochondria can be controlled at the single organelle level to ultimately support cell viability and function. It is also worth noting that cells in the LP and IEL fraction of the gut showed a reduced ATP:AMP ratio, which may affect their long-term performance in coping with the continuously challenging environment of the gut.

Our metabolomic analysis of the interstitial fluid showed that in steady-state physiological conditions PGE$_2$ is characteristic of the gut microenvironment and affects CD8$^+$ T cell mitochondrial content, leaving other cells, such as macrophages, unaffected. PGE$_2$ has been widely recognized to affect several aspects of T cell immune responses and intestinal physiology[41,58−62]. Indeed, PGE$_2$ is an integral component of the gut environment, essential to promote epithelial maintenance upon the constitutive stress posed by the microflora[41,42]. While gut IEL have been shown to have reduced mitochondrial mass as well as a different proteome as compared to mLN-resident CD8$^+$ T cells[34,63], our data establish a mechanistic link between physiological PGE$_2$ concentrations in the intestine and how they uniquely affect T cells to shape their metabolic profile and adaptation to the gut environment. It remains to be assessed how PGE$_2$ production is regulated in the gut and whether PGE$_2$ concentrations increase in other peripheral tissues upon pathological changes, such as described in the tumor microenvironment[64]. Similarly, as sensing of PGE$_2$ is affected by expression of EP receptors, the tissue-specific receptor expression likely influences the effect of PGE$_2$ in different cell types. Differently from CD8$^+$ T cells, gut LP-isolated macrophages did not reduce their mitochondrial content. While a subset of gut-resident macrophages, as well as other intestinal cells, was found to express *Ptger4* and/or other EP receptors[65−67], it is likely that additional factors play a role in shaping the mitochondrial content of gut-isolated cells. For instance, we showed that the activation profile of macrophages could circumvent the mitochondrial changes driven by PGE$_2$ exposure[43]. While our data clearly highlight how the tissue microenvironment impinges on the metabolic fitness of immune cells, the mechanistic link between the

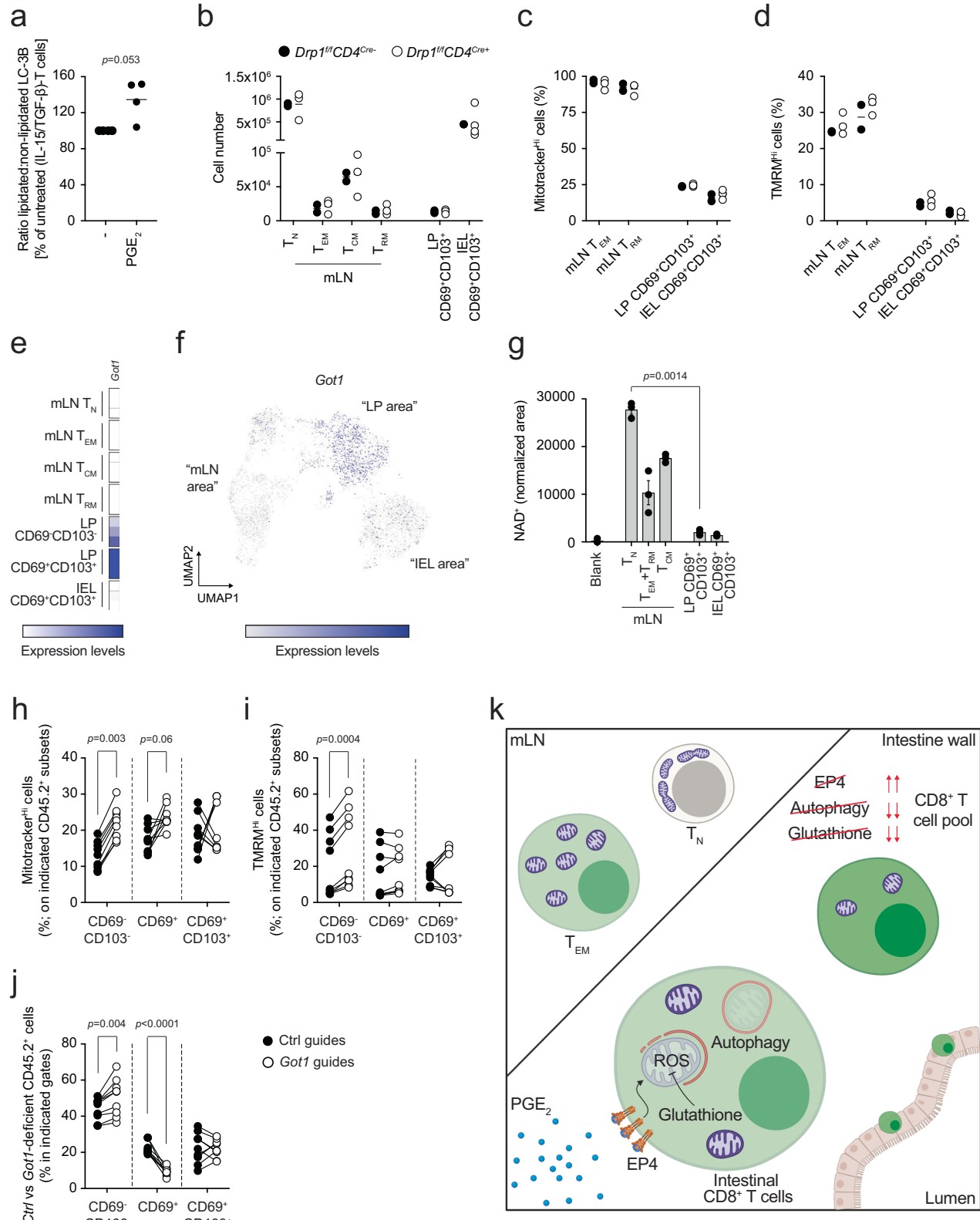

sensing of PGE$_2$, the reduction of mitochondrial content, and the acquisition of the *bona fine* resident memory phenotype is yet to be defined. Tracing experiments of antigen-specific cells across time (from naïve to memory time points) and space (secondary lymphoid tissues to small intestine) will help addressing this point. While PGE$_2$ itself was found to trigger autophagy, this could also occur as a consequence of the PGE$_2$-driven mitochondrial depolarization. The

binding of PGE$_2$ to EP4 engages the cAMP/PKA pathway, and it has been suggested that PGE$_2$ may directly activate PI3K[68], providing a potential explanation of the effect of the PI3K inhibitor wortmannin in preventing both mitochondrial clearance and reduction of mitochondrial membrane potential, as compared to bafilomycin A1, an inhibitor of late stages of autophagy, the effect of which is restricted to mitochondrial clearance. Our data implicating a role for autophagy in

**Fig. 5 | Got1 links PGE₂ with the regulation of mitochondrial content in intestinal LP and epithelial CD8⁺ T cells. a** Western blot of LC3-I and LC3-II in (IL-15/TGF-β)-T cells cultured for 6 days and treated for 24 h with or without 100 nM PGE₂. Dot plot shows quantification by densitometry. Lines in dot plots show mean values. Dot plots show cumulative data of $n = 4$ biological replicates over two independent experiments. Statistics were performed using two-tailed paired $t$ test. **b** Cell numbers of CD8⁺ T cells subsets in mLN, LP and IEL of *Drp1^{fl/fl}CD4Cre⁺* and *Drp1^{fl/fl}CD4Cre⁻* mice. Lines in the dot plot show mean values and the data show $n = 2$–4 biological replicates over two independent experiments. **c, d** Mitotracker green and TMRM in cells from mLN and intestine of *Drp1^{fl/fl}CD4Cre⁺* and *Drp1^{fl/fl}CD4Cre⁻* mice. Lines in the dot plot show mean values and the data show $n = 2$–4 biological replicates over two independent experiments. **e** Heatmaps show the expression of *Got1* in bulk RNA sequencing data obtained from the indicated cells. Color-coding is as per legend (white, low expression; blue, high expression) and shows relative values within the *Got1* gene. Data show three independent experiments. **f** UMAP of the single cell RNA and ADT sequencing data showing expression across different clusters of *Got1*. **g** Mass spectrometry quantification of NAD⁺ in the indicated cells. The bars show mean ± SEM. Data show three independent

experiments analyzed simultaneously. Statistics were performed using one-way ANOVA and Tukey's multiple comparison correction. Mitotracker green (**h**) and TMRM (**i**) in Ctrl vs *Got1*-deleted CD69⁻CD103⁻, CD69⁺ and CD69⁺CD103⁺ cells isolated from LP of mice orally challenged with *LmOVA*. Lines in the dot plot show pairing within single mice, and dot plots show cumulative data of $n = 9$ biological replicates over two independent experiments. **j** Distribution of Ctrl vs *Got1*-deleted CD8⁺ T cells gated in the population of CD45.2⁺ cells between CD69⁻CD103⁻, CD69⁺ and CD69⁺CD103⁺ populations, upon *LmOVA* challenge in LP. Lines in the dot plot show pairing within single mice; dot plots show cumulative data of $n = 9$ biological replicates over two independent experiments. **k** After activation CD8⁺ T cells migrate and enter the intestine. Here, they sense PGE₂ via the EP4 receptor. PGE₂ drives the reduction of mitochondrial membrane potential, in part via alteration of the malate-aspartate shuttle, and leads to drop in mitochondrial mass. Autophagy contributes to the clearance of depolarized mitochondria, whereas glutathione maintains the redox balance by scavenging mitochondrial ROS, to ultimately shape the pool of CD8⁺ T cells. Image created with Biorender.com. **h**–**j** Statistics were performed using two-way ANOVA and Sidak's multiple comparison correction. Source data are provided as a Source Data file.

mitochondrial quality control of LP and IEL CD8⁺ T cells are in line with findings that highlighted autophagy as a key pathway required for the establishment of CD8⁺ T cell tissue residence[69]. In addition, while the mitochondrial fission-inducing protein DRP1 is required for mitophagy in cardiomyocytes[70], deletion of *Drp1* did not prevent the loss of mitochondrial mass observed upon entry of CD8⁺ T cells in the intestine. It is possible that gut-adapting CD8⁺ T cells use additional strategies beyond autophagy to clear mitochondria. For example, migratory cells are able to dispose of damaged mitochondria through a process called mitocytosis[71].

While deletion of the EP4 receptor promoted the accumulation of CD8⁺ T cells in the intestinal LP and epithelial layer, tampering with autophagy or glutathione pathways negatively affected the establishment of the T cell pools in the analyzed intestinal compartments, supporting a key role for these processes in regulating T cell function and intestinal homeostasis[48,49,72–75]. Analysis of bona fide memory time points will clarify whether PGE₂ sensing, autophagy and glutathione are involved in the establishment of the CD8⁺ T cell memory compartment in the intestine. Of note, the teleology of our findings may appear peculiar, that is, why LP- and IEL-isolated CD8⁺ T cells express a receptor (EP4) that hinders their fitness. It is, however tempting to speculate that PGE₂ may play a role in 'updating' the CD8⁺ T cell pool. By triggering mitochondrial dysfunction, and thus shortening the life span of the cells[76], PGE₂ may continuously remodel the CD8⁺ T cell repertoire to adapt to ever-changing conditions, such as those presented by the microbiota or food antigens. Alternatively, PGE₂ sensing and the subsequent reduction of mitochondrial content in T cells, may represent one of the strategies in place in the gut to constrain the CD8⁺ immune response mediated by $T_{EM}$. Our results seem to support this hypothesis as *Ptger4* deletion not only triggered accumulation of CD8⁺ T cells in LP and IEL, but also retained them in the CD69⁻CD103⁻ status. While autophagy is intrinsically required to clear the damaged mitochondria and preserve a viable size of the CD8⁺ T cell compartment, elevated glutathione may be a consequence of the depolarization of mitochondria and elevated mitochondrial ROS driven by PGE₂, and could modulate its 'purging' effect on the T cell pool. In addition, our findings may aid in the interpretation of the counterintuitive clinical phenomenon of non-steroidal anti-inflammatory drug (NSAID)-induced colitis (NSAIDs are known inhibitors of PGE₂ biosynthesis), suggesting a possible role for the accumulation of CD8⁺ T cells in driving the disease[77]. It is however worth noticing that under certain experimental settings, T cells have been shown to be dispensable for the induction of NSAID-triggered colitis[59]. Finally, our data showed that gut LP and epithelium localization of CD8⁺ T cells is characterized by reduced mitochondrial content, as well as by reduced levels of NAD⁺, and a possibly diminished glycolytic flux. The origin of these phenomena likely lies in the effect of PGE₂ on

the function of the malate-aspartate shuttle, causing the "cooling down" of CD8⁺ T cell metabolism upon sensing the gut microenvironment, a hypothesis in line with recent literature showing limited metabolic activation of gut-resident T cells[78].

Immune checkpoint inhibitors (ICI) have thus far shown limited success in colorectal cancers (CRC)[79]. We speculate that mitochondrial loss in T cells could be an underlying factor contributing to the lack of efficacy of ICI in CRC. While exposure to PGE₂ does not appear to diminish T cell function, such as the production of IFN-γ, it nevertheless limits cell numbers in the niche. Moreover, recent data showed that inhibition of PGE₂ sensing by EP2 and EP4 receptors promoted the accumulation of CD8⁺ T cells within tumors[80]. Our data corroborate these findings by showing that EP4-deficient T cells outcompete wild-type cells to occupy the LP and IEL niche in greater numbers, suggesting that pharmacological blockade of the PGE₂ receptor or PGE₂ synthesis might increase T cell numbers in the gut. We question whether combining ICI with PGE₂ pathway inhibition could enhance the outcome of ICI in CRC by increasing the number of CD8⁺ T cells in the niche, as previously suggested[64]. Further, it is likely that ICI efficacy may require T cells to have "fit" mitochondria, and not dysfunctional ones, as result from PGE₂ exposure. We speculate that the PGE₂-autophagy-glutathione pathway could be targeted to fine-tune the accumulation of CD8⁺ T cells in the gut, to modulate immune responses underscoring inflammatory bowel disease and cancer.

In summary, we propose that a balance between PGE₂ sensing, mitochondrial clearance, and glutathione-regulated redox balance shapes the adaptation of CD8⁺ T cells to the gut microenvironment and ultimately the establishment of the CD8⁺ T cell pool. PGE₂ is required for gut epithelial stability and repair, therefore, it is an integral part of this tissue's environmental signature. As T cells transition from the mLN to the gut LP and epithelial layer, they become exposed to PGE₂. This causes mitochondrial depolarization and affects mitochondrial fitness. However, T cells adapt by engaging autophagy to clear depolarized mitochondria and augment glutathione synthesis to regulate cellular redox balance—these metabolic adaptations allow gut T cells to maintain function and carry on. Thus, we conclude that the environment surrounding CD8⁺ T cells impinges on their metabolic choices, complements cytokine sensing and transcriptional rewiring, and allows T cells to adapt and function in the newly encountered tissue niche.

## Methods

### Mice, in vivo treatments and ethical statement

C57BL/6J mice (RRID: IMSR_JAX:000664), PhAM mice (RRID: IMSR_JAX:018397), CD4^{Cre} mice (RRID: IMSR_JAX:017336), Drp1^{floxed} mice (gift from Hiromi Sesaki), Great mice (interferon-γ reporter, RRID: IMSR_JAX:017580), major histocompatibility complex (MHC) class

I-restricted ovalbumin (OVA)-specific TCR OT-I transgenic mice (RRID: IMSR_JAX:003831), CD45.1 congenic mice (RRID: IMSR_JAX:002014), Thy1.1 congenic mice (RRID: IMSR_JAX:000406) were purchased from The Jackson Laboratory. Mito-roGFP2-Orp1 mice (tracking mitochondrial $H_2O_2$ production)[46,47] were a kind gift from Tobias P. Dick (DKFZ).

All mice were maintained in the animal facilities at the Max Planck Institute for Immunobiology and Epigenetics, under specific-pathogen free (SPF) conditions and following institutional animal use and care guidelines. Mice were exposed to a 14 h/10 h light/dark cycle and fed ad libitum with acidified water (pH 2.5–3.3). Euthanasia and animal procedures were conducted on 8–12 weeks old male and/or female mice, age-matched and sex-matched, in compliance to § 4, paragraph 3 of the German Animal Protection Act, animal licenses 35-9185.81/G-20/107 and 35-9185.81/G-20/101, approved by the Regierungspräsidium Freiburg.

To assess the residency of T cells in various tissues, mice were injected intravenously (iv) with 2.5 µg of fluorescently labeled anti-CD45 antibody (clone 30-F11), 10 min before euthanasia, perfusion and analysis.

To establish the antigen-specific *Listeria monocytogenes* infection model, we proceeded as follows. $10^4$ naive congenically-marked OT-I cells (CD90.1+ in experiments in Fig. 2d, f, CD45.2+ in experiments in Figs. 3–5 and Supplementary Figs. 4–7) were adoptively transferred in unchallenged recipient CD90.2+ CD45.1+ mice. In the competition experiments shown in Figs. 3–5, $10^4$ or $10^5$ (*Got1* experiments) naive congenically-marked OT-I cells (CD45.2+ and CD90.2+) and $10^4$ or $10^5$ (*Got1* experiments) naive congenically-marked OT-I cells (CD45.2+ CD90.1+) were genetically modified by CRISPR-Cas9 (see details below) and co-transferred in unchallenged recipient CD45.1+ mice. 24 h after the adoptive transfer, recipient mice were challenged with $10^9$ CFU of *LmOVA* intra-gastric (oral gavage, og). Mice were monitored daily, euthanized and analyzed 7 or 24 days post-infection.

## Isolation of cells
To isolate cells from the spleen and mesenteric lymph nodes (mLN), the organs were mashed on a 70 µm strainer in medium containing RPMI1640, 10% heat-inactivated fetal calf serum (FCS), 4 mM glutamine, 1% penicillin/streptomycin solution and 55 µM β-mercaptoethanol (hereafter named complete medium). Cell suspensions were treated with Gey's solution for 3 min if red blood cell lysis was required. Finally, cells were extensively washed and suspensions used for downstream applications. Purification of CD8+ T cells for sorting or cell culture was performed using a mouse CD8+ T cell isolation kit (Stem Cell Technologies, Cat# 19853), following the manufacturer's instructions.

Isolation of cells from the small intestine was performed as follows. Upon harvesting, fat tissue bordering the small intestine was carefully removed and Peyer's patches were excised. Small intestines were cut open, washed with ice-cold PBS 1X + 25 mM Hepes, cut into 2 cm-long pieces and collected in medium containing RPMI1640, 25 mM Hepes, 3% FCS, 4 mM glutamine, 1% penicillin/streptomycin and 55 µM β-mercaptoethanol (hereafter named 3% FCS medium). Small intestines were incubated in 3% FCS medium added with 5 mM EDTA and 1 mM dithiothreitol (DTT) for 25 min, at 37 °C, 5% $CO_2$ and agitation (1st incubation). After the incubation, cells in suspension were collected, consecutively strained through 100 µm and 40 µm strainers, washed and kept as intraepithelial lymphocyte (IEL) fraction. The leftover small intestine pieces were thoroughly (4 times) washed by shaking with ice-cold medium containing RPMI1640, 25 mM Hepes, 4 mM glutamine, 1% penicillin/streptomycin and 55 mM β-mercaptoethanol (hereafter named serum-free medium) added with 2 mM EDTA. Washed small intestine pieces were then finely chopped and incubated in serum-free medium added with 100 µg/ml Liberase TL (Roche) and 50 µg/ml DNAse I (Roche) for 35 min, at 37 °C, 5% $CO_2$ and agitation (2nd incubation). After digestion, suspensions were strained through 70 µm strainers, washed and kept as lamina propria

(LP) fraction. LP and IEL were enriched for leukocytes using a three-layers percoll gradient (percoll 75%, 40% and 30%) centrifuged for 20 min, at room temperature, 680 g, without any centrifuge acceleration or brake. The leukocyte layer between percoll 75% and 40% was collected as enriched LP or IEL fraction, extensively washed and suspensions used for downstream applications. In some experiments, 1 µM of KN-62 (Sigma), inhibitor of P2RX7 used to limit cell death during intestine digestion[37], was added to media used in the 1st and 2nd incubation.

T cell isolation from the liver was performed as follows. Whole liver was collected in ice cold complete medium, finely chopped and incubated in RPMI1640 added with 1 mg/ml collagenase IA (Sigma) and 50 µg/ml DNAse I (Roche) for 45 min, at 37 °C, 5% $CO_2$ and agitation. After digestion, suspensions were strained through 70 µm strainers, washed and enriched for leukocytes using percoll 33% and centrifuging them for 20 min, at room temperature, 680 g. The leukocyte pellet was collected, extensively washed, red blood cell lysis was performed and suspensions used for downstream applications.

Lung digestion was performed as follows. Lungs were collected in ice-cold complete medium, finely chopped and incubated in complete medium added with 1 mg/ml collagenase IA (Sigma) and 50 µg/ml DNAse I (Roche) for 45 min, at 37 °C, 5% $CO_2$ and agitation. After digestion, suspensions were strained through 70 µm strainers, washed and enriched for leukocytes using a three-layers percoll gradient (percoll 75%, 40% and 30%) centrifuged for 20 min, at room temperature, 680 g, without any centrifuge acceleration or brake. The leukocyte layer between percoll 75% and 40% was collected, extensively washed and suspensions used for downstream applications.

To isolate human PBMCs from fresh blood samples, blood was first diluted 1:1 with PBS 1X + 2% FCS. Samples were then layered in SepMate™ PBMC isolation tubes (Stem Cell Technologies) preloaded with Lymphoprep (Stem Cell Technologies) and centrifuged for 10 min, at room temperature, 1200 g. The PBMC layer was collected and extensively washed in PBS 1X + 2% FCS and suspensions used for downstream applications.

For isolation of cells from human gut biopsies, we proceeded as follows. Fresh or frozen gut biopsies (5 mm; matched with fresh or frozen PBMCs) were mashed through 70 µm strainers in complete medium. Suspensions were extensively washed and finally used for downstream applications.

## Fluorescence-activated cell sorting (FACS)
To analyze the CD8+ T cell response in humans, we used the surface markers CD45RA, CD27, CD69 and CD103, identifying by flow cytometry five CD8+ T cell populations: CD45RA+CD27+ $T_N$, CD45RA−CD27− $T_{EM}$, CD45RA+CD27- $T_{EMRA}$ and CD45RA−CD27+ $T_{CM}$ isolated from blood; CD69+CD103+ $T_{RM}$ isolated from gut biopsies. To analyze the continuum of the gut immune response in mouse we used the surface markers CD62L, CD44, CD69 and CD103, identifying by flow cytometry seven CD8+ T cell populations that approximate the transition of CD8+ T cells from mLN to the intestine: CD62L+CD44− $T_N$, CD62L+CD44+ $T_{CM}$, CD62L−CD44+CD69−CD103− $T_{EM}$ and CD62L−CD44+CD69+CD103+ $T_{RM}$ isolated from mLN; CD44+CD69−CD103- and CD44+CD69+CD103+ from LP; and CD44+CD69+CD103+ induced CD8αβ cells from IEL.

Before staining with MitoTracker™ Green or TMRM, cell suspensions were counted and equal cell numbers were stained to prevent uneven staining of suspensions with different cell numbers, and thus bias the analysis. The following dyes and staining procedures were used. MitoTracker™ Green FM (Invitrogen) was used to stain cells at a concentration of 100 nM in complete medium, for 30 min, at 37 °C, 5% $CO_2$. In some experiments, 50 µM of Verapamil (Sigma), inhibitor of mitochondrial efflux pumps encoded by *Bcrp1* and *Mdr1a/b*, used to prevent efflux of MitoTracker™ green FM from mitochondria[36], was added to the staining medium. Tetramethylrhodamine methyl ester perchlorate (TMRM, Invitrogen) was used to stain cells at

concentrations of 50 or 10 nM, according to the experiment, in complete medium, for 30 min, at 37 °C, 5% CO₂. In experiments where TMRM kinetic was analyzed, 500 nM cyclosporin H (Sigma), another inhibitor of mitochondrial efflux pumps used to prevent efflux of TMRM from mitochondria, was added to the staining and acquisition media. In the same experiments, oligomycin (Sigma), inhibitor of complex V of the electron transport chain (ETC) was used at a concentration of 10 μM, whereas carbonilcyanide p-triflouromethoxy-phenylhydrazone (FCCP, an ionophore that permeabilize the inner mitochondrial membrane to H⁺, Sigma) was used at a concentration of 10 μM. CellROX™ Deep Red or Green (Invitrogen) was used to stain cells at a concentration of 250 nM in complete medium, for 30 min, at 37 °C, 5% CO₂. In the experiments using Mito-roGFP2-Orp1 mice, gates were set upon treating the cells with the reducing agent DTT (10 mM, 15 min) or the oxidizing reagent diamide (1 mM, 15 min).

Cell viability was quantified by flow cytometry using LIVE/DEAD™ blue or near-IR dyes (Invitrogen) or Zombie Green™ dye (Biolegend), following manufacturer instructions.

Surface staining was performed in PBS 1X + 2% FCS + 5 mM EDTA, for 20 min, on ice, in the dark, upon blocking cells with anti-CD16/32 (clone: 93, Biolegend). The following antibodies (clone in brackets) were used for surface staining of murine cells: anti-CD45 (30-F11), anti-CD8α (53-6.7), anti-CD8β (53-5.8), anti-CD62L (MEL-14), anti-CD44 (IM7), anti-CD69 (H1.2F3), anti-CD64 (X54-5/7.1), anti-F4/80 (BM8), anti-Ly6G (1A8), anti-TIM4 [54(RMT4-54)], anti-CD45.2 (104), anti-Thy1.1 (OX-7) all from Biolegend; anti-CD103 (2E7), from Invitrogen; anti-Siglec F (E50-2440), from BD. The following antibodies (clone in brackets) were used for surface staining of human cells: anti-CD19 (HIB19), anti-CD3 (HIT3a), anti-CD8 (HIT8a), anti-CD69 (FN50), anti-CD103 (Ber.ACT8), anti-CD45RA (HI100), anti-CD27 (LG3A10), all from Biolegend.

Intracellular staining was performed upon restimulation of ex vivo isolated cells in U-bottom plates coated with anti-CD3 at 5 μg/ml and resuspending the cells in complete medium added with 100 U/ml IL-2 and soluble αCD28 at 0.5 μg/ml, in presence of brefeldin A 1:1000 (Biolegend), under 5% CO₂, atmospheric oxygen, at 37 °C in a humidified incubator for 4 h. After surface staining, intracellular staining was performed using Citofix/Cytoperm™ kit (BD) following manufacturer instructions and using the following antibodies: anti-interferon γ (XMG1.2), anti-TNF (MP6-XT22), all from Biolegend.

Cells were acquired on BD LSRFortessa™ or BD FACSymphony™, or they were sorted using BD FACSAria™. Raw data were analyzed with FlowJo (BD).

## TotalSeq™ antibody staining

TotalSeq™-A Custom Mouse panel including 198 antibodies was obtained from Biolegend (#99833). The staining procedure was performed immediately after cell sorting and strictly following manufacturer instructions. Specifically, 25 μl of TotalSeq™-A antibody cocktail were used to label 25 μl of cell suspension containing 5 × 10⁵ cells. Four groups of samples have been prepared for TotalSeq™-A antibody staining and the following single cell RNA sequencing (scRNAseq), in biological triplicate: 1. naive and central memory CD8⁺ T cells (CD62L⁺) isolated from mLN (mLN T_N and T_CM); 2. effector memory CD8⁺ T cells (CD62L⁻ CD44⁺) isolated from mLN (mLN T_EM); 3. total CD8⁺ T cells isolated from LP (LP CD8); 4. total CD8⁺ T cells isolated from IEL (IEL CD8). This strategy was designed to focus the analysis on CD8⁺ T cells (hence the cell sorting approach), to monitor the entire course of the response (from mLN to IEL) and to enhance the resolution of the analysis to the stages involving the T cell adaptation to the intestinal tissue (hence the higher number of cells analyzed from LP).

## Single-cell RNA sequencing

After staining with TotalSeq™-A antibodies, suspensions were used for single-cell RNA sequencing (scRNAseq), hereafter briefly described.

Following the protocol CG000204 RevD from 10X Genomics, we aimed at analyzing 1000 cells of group 1 (mLN T_N and T_CM), 5000 cells of group 2 (mLN T_EM), 10,000 cells of group 3 (LP CD8) and 5000 cells of group 4 (IEL CD8). At step 2.2, we followed Biolegend instructions to add ADT additive primers. Step 2.3 (and only this step) was performed following the protocol CG000185 RevD from 10X Genomics. 3′ gene expression libraries were further processed following the protocol CG000204 RevD from 10X Genomics, whereas ADT libraries (cell surface protein libraries) were processed following instructions from Biolegend. 3′ gene expression and ADT libraries were quantified by fragment analyzer and pooled for multiplexed sequencing on a NovaSeq 6000 instrument by the Deep-sequencing Facility at the Max Planck Institute for Immunobiology and Epigenetics. 3′ gene expression libraries were run with a read length of 2 × 100 whereas ADT libraries were run with a read length of 2 × 50. Both libraries were sequenced using the following parameters, as recommended by 10X Genomics and Biolegend: Read 1: 28 cycles, i7: 8 cycles, i5: 0 cycles, Read 2: 91 cycles. 50,000 reads/cell were sequenced for 3′ gene expression libraries, 12,500 reads/cell were sequenced for ADT libraries.

Samples were demultiplexed and aligned using Cell Ranger 6.1 (10X Genomics) to genome build mm10 to obtain a raw read count matrix of barcodes corresponding to cells and features corresponding to detected genes. Read count matrices were processed, analyzed and visualized in R v. 4 (R Core Team 2013) using Seurat v. 4[81] with default parameters in all functions, unless specified. Poor quality cells, with low total unique molecular identifier (UMI) counts (<4000), low number of detected genes (<1500) and high percent mitochondrial gene expression (>20%), were excluded. Filtered samples were normalized using a regularized negative binomial regression (SCTransform)[82] and merged. Principal component analysis was performed on merged matrices and the first 50 principal components were used for downstream analysis. Integrated gene expression matrices were visualized with a UMAP[83] as a dimensionality reduction approach. For ADT libraries, samples were demultiplexed and then processed with CITE-seq-Count v. 1.4.5[84]. Later merged and harmonized to remove biological replicate associated effects using harmony[85]. A combined UMAP for RNA and ADT profiles was generated by building a weighted nearest neighbors graph based on their respective PCA projections[81]. Resolution for cell clustering was determined by evaluating hierarchical clustering trees at a range of resolutions (0–1.2) with Clustree[86], selecting a value inducing minimal cluster instability. Differentially expressed genes between clusters were identified as those expressed in at least 25% of cells with a greater than +0.5 log₁₀ fold change and an adjusted p value of less than 0.01, using the FindMarkers function in Seurat v.4 with all other parameters set to default. Ribosomal protein genes were excluded from results. Cluster specific genes were explored for pathway enrichment using Biological Process Gene Ontology annotation with goseq v. 1.40.0[87]. Gene set scores were calculated using the AddModuleScore function in Seurat v.4 with default parameters. Briefly, the average expression levels of each identified gene set were calculated on a single cell level and subtracted by the aggregated expression of randomly selected control gene sets. For this purpose, target genes are binned based on averaged expression, and corresponding control genes are randomly selected from each bin. Cell trajectories and pseudotime estimations were calculated with Slingshot v. 1.6.1[35], using UMAP projection and pre-calculated clustering as input for getLineages and getCurves functions with default parameters, setting origin to cluster 6. Trajectory-dependent gene regulation was visualized by fitting general additive models[79] using gam v. 1.20 (Hastie, T 2020) using locally estimated scatterplot smoothing (loess) smooth terms.

## Western blot

For western blotting analysis, cells were washed with ice-cold PBS 1X and lysed in 1X Cell Signaling lysis buffer (20 mM Tris-HCL [pH 7.5], 150 mM NaCl, 1 mM $Na_2EDTA$, 1 mM EGTA, 1% Triton X-100, 2.5 mM sodium pyrophosphate, 1 mM β-glycerophosphate, 1 mM $Na_3VO_4$, 1 μg/mL leupeptin), supplemented with 1X Protease Inhibitor Cocktail (Cell Signaling, #5871) for 30 min, on ice, followed by centrifugation at 12,000 $g$ for 10 min, at 4 °C. Cleared protein extracts were denatured with NuPAGE™ LDS loading buffer (Invitrogen) added with 50 mM DTT for 10 min at 75 °C and loaded onto precast NuPAGE™ 4–12% bis-tris protein gels (Invitrogen), or NuPAGE™ 12% bis-tris protein gels (these ones were used for LC3 immunoblotting, Invitrogen). Samples were loaded according to the cell number: Between $5 \times 10^4$ and $2 \times 10^5$ cells per sample were analyzed, normalizing the loading within every experiment. Protein samples were run using MES buffer 1X (Invitrogen). Proteins were transferred onto nitrocellulose membranes using the iBLOT2 system (Invitrogen) following the manufacturer protocols. Membranes were blocked for 1 h with 5% w/v milk in TBS 1X added with Tween-20 0.05% and incubated with primary antibodies in 5% w/v milk in TBS 1X added with Tween-20 0.05% overnight at 4 °C or 1 h at room temperature. The following primary antibodies were used: anti-TOM20 (clone D8T4N, 1:1000, Cell Signaling), anti-TFAM (polyclonal, 1:1000, Abcam), anti-β actin (clone 13E5, 1:10,000, Cell Signaling), anti-OPA1 (clone 18/OPA-1, 1:2000, BD), anti-EP4 (encoded by *Ptger4*, clone C-4, 1:100, Santa Cruz), anti-ATG5 (clone D5F5U, 1:1000, Cell Signaling), anti-Gclc (clone OTI1A3, 1:1000, Invitrogen), anti-LC3B (polyclonal, 1:3000, Sigma), anti-SQSTM1/p62 (clone D1Q5S, 1:1000, Cell Signaling), anti-Got-1 (clone E4A4O, 1:1000, Cell Signaling). All primary antibody incubations were followed by incubation with secondary HRP-conjugated antibodies (Pierce) in 5% w/v milk in TBS 1X added with Tween-20 0.05% for 1 h at room temperature and visualized using SuperSignal West Pico or Femto Chemiluminescent Substrate (Pierce) using Biomax MR Film (Kodak) or a Bio-Rad ChemiDoc™ Imaging System. Optical density of the signals on the film was quantified using grayscale measurements in ImageJ software (NIH) and converted to fold change, normalized to the loading control.

## Fluorescence microscopy

Spinning disk confocal imaging was performed as follows. Cells were isolated and sorted as previously described from unchallenged PhAM mice. A minimal surface staining panel was used for sorting the cells to minimize interference with the PhAM signal during spinning disk acquisition. After sorting, cells were resuspended in complete medium and transferred to Lab-Tek™ II chambers (Nunc) coated with poly-D-lysine. Chambers were spun to favor the attachment of the cells to the bottom of the chambers. During the entire experiment, cells were maintained alive at 37 °C, 5% $CO_2$ using a Tokai Hit chamber. Fluorescence of Dendra2 (fluorescent protein in PhAM mice) and Hoechst 33342 (nuclear counterstaining) was acquired using a Zeiss spinning disk confocal with an Evolve (EMCCD) camera. Confocal images were deconvoluted using Huygens software and analyzed using Imaris imaging software.

## Transmission electron microscopy

Between $2.5 \times 10^5$ and $10^6$ cells were processed for electron microscopy. After FACS sorting, cells were fixed in glutaraldehyde 2.5% in 0.1 M sodium cacodylate buffer pH 7.4, for 20′ at RT, washed in the same buffer, and stored at +4 C until analysis.

## Bulk RNA sequencing

For bulk RNA sequencing, $10^5$ cells per target population from each of three biological replicates were isolated, processed and analyzed. RNA was extracted using RNAqueous™ Total RNA isolation kit (Ambion) according to manufacturer instructions and quantified using Qubit 2.0 (Thermo Fisher Scientific) following manufacturer instructions.

Libraries were prepared using the NEBNext Single Cell/Low Input RNA Library Prep Kit for Illumina and sequenced with a NovaSeq 6000 instrument by the Deep-sequencing Facility at the Max Planck Institute for Immunobiology and Epigenetics. Sequenced libraries were processed with deepTools[88,89], using STAR[90], for trimming and mapping, and featureCounts[91] to quantify mapped reads. Raw mapped reads were processed in R (Lucent Technologies) with DESeq2[92], to determine differentially expressed genes (adjusted $p$ value < 0.05, $log_2$ fold change > 1) and generate normalized read counts to be visualized as heatmaps using Morpheus (Broad Institute).

## Human samples and ethical statement

Biopsies from $n = 5$ healthy donors presenting for routine endoscopy were included in the study. None of the included biopsies showed macroscopic signs of active inflammation, and patients were asymptomatic. Human participants were recruited from the University Hospital Freiburg in a sequential manner with every patient undergoing endoscopy on the days of recruitment, being approached for participation in the study. Given no pre-selection based on age, gender, medical history (except for gastrointestinal diseases), or any other criteria, we do not perceive any bias in the recruitment of individuals. Blood and intestinal tissue were collected after obtaining the written consent of all study participants and following approval of the Institutional Review Boards (Ethics Committee of the Albert Ludwigs University, Freiburg; #407/16 and #14/17). The study was performed in agreement with the principles expressed in the Declaration of Helsinki (2013).

## Isolation of interstitial fluid

For isolation of tissue interstitial fluid we proceeded as follows adapting an already published protocol[93]. Upon perfusion with ice cold PBS 1X, tissues (spleen, mLN, small intestine, liver and lungs) were carefully excised to maintain their integrity and collected in ice-cold PBS 1X. Blood was withdrawn by cardiac puncture and maintained on ice in absence of anticoagulants. Fecal content was collected from the lumen of the ileum section of the small intestine and kept on ice. Coagulated blood was spun for 10 min, 5000 $g$, at 4 °C and serum was collected and kept on ice until further processing. Fecal content was spun for 10 min, 400 $g$, at 4 °C and supernatant was collected and kept on ice until further processing. Tissues were gently tapped on adsorbing paper, wrapped in a 20 μm nylon mesh and placed in a 1.5 ml tube. Tubes were centrifuged for 5 min, 70 $g$, 4 °C to remove leftover PBS 1X. Tissue wrapped in the nylon mesh was transferred to another 1.5 ml tube and centrifuged for 10 min, 400 $g$, 4 °C. The flow-through is tissue interstitial fluid and volumes between 1 and 15 μl were collected. Interstitial fluid was kept on ice until further processing.

## Metabolomic analysis

Untargeted metabolomic analysis of interstitial fluid was performed as follows. Nine volumes of ice-cold methanol 100% were added to the interstitial fluid, the samples vortexed and cooled on dry ice. Samples were centrifuged for 3 min, 20,000 $g$, at −9 °C. Supernatant was collected and samples were stored at −80 °C until further processing following protocols previously described[94]. Metabolites were analyzed and identified by fragmentation and retention time using Metaboscape software (Bruker).

Targeted metabolomic analysis was performed as follows. After sorting, $2 \times 10^5$ cells were resuspended in complete medium and rested for 30 min at 37 °C, 5% $CO_2$. Cells were washed in ice cold 3% glycerol and the pellet resuspended in extraction solution (ice-cold 80% methanol). Samples were incubated for 5 min on ice and centrifuged for 3 min, 20,000 $g$, at −9 °C. Supernatant was collected and kept on dry ice. Cell pellets were further resuspended for two more times with extraction solution and treated as previously described to further

collect supernatants. Samples were finally dried using a Genevac EZ2 speed vac and stored at −80 °C until further processing following protocols previously described[94]. Targeted metabolite quantification by LC-MS was carried out using an Agilent 1290 Infinity II UHPLC in line with an Agilent 6495 QQQ-MS operating in MRM mode. MRM settings were optimized separately for all compounds using pure standards. LC separation was on a Phenomenex Luna propylamine column (50 × 2 mm, 3 μm particles) as described previously[94]. Briefly, a solvent gradient of 100% buffer B (5 mM ammonium carbonate in 90% acetonitrile) to 90% buffer A (10 mM $NH_4$ in water) was used. Flow rate was from 1000 to 750 μL/min. Autosampler temperature was 5 °C and injection volume was 2 μL. Raw data were analyzed using the R package automRm[95]. Further analysis was performed using MassHunter software (Agilent).

### In vitro T cell culture and treatments

Isolated CD8+ T cells from OT-I mice were maintained in the naive status after CRISPR-Cas9 gene targeting by resting them overnight in complete medium added with IL-7 (Peprotech) at 10 ng/ml. Isolated T cells were otherwise activated using plate-bound anti-CD3 (5 μg/ml, inVIVOMAb Cat# BE0002) in complete medium added with 100 U/ml IL-2 (Peprotech) and soluble αCD28 (0.5 μg/ml, inVivoMab Cat# BE0015) under 5% $CO_2$, atmospheric oxygen, at 37 °C in a humidified incubator. At day 2 post activation, and then daily up to day 6, media were refreshed and cells counted and replated at a density of $10^6$ cells/ml [for (IL-2)-T cells] or $1.5 \times 10^6$ cells/ml [for (IL-15/TGF-β)-T cells]. (IL-2)-T cells were maintained in 100 U/ml IL-2 throughout the culture; to generate (IL-15/TGF-β)-T cells, cells were cultured in 10 ng/ml IL-15 and 10 ng/ml TGFβ starting from day 3 post-activation. Where indicated cells were treated with vehicle control (0.1% ethanol), $PGE_2$ (Sigma) at the indicated concentrations and for the indicated time, $PGE_1$ (Sigma), $PGF_{2α}$ (Sigma), $PGI_2$ sodium salt (Sigma) and 15-deoxy-Δ12-14 $PGJ_2$ (Sigma) at 100 nM, bafilomycin A1 (Sigma) at 100 nM, Wortmannin (PI3K inhibitor, Selleckchem) at 10 μM and SBI-0206965 (ULK1 inhibitor, Sigma) at 2 μM.

### CRISPR-Cas9 gene targeting

The following gRNA were used to target the genes *Ptger4*, *Atg5*, *Gclc*, and *Got1* in naive CD8+ T cells: *Ptger4* (Mm.Cas9.PTGER4.1.AA; Mm.Cas9.PTGER4.1.AG), *Atg5* (Mm.Cas9.ATG5.1.AA; Mm.Cas9.ATG5.1.AB), *Gclc* (Mm.Cas9.GCLC.1.AC; Mm.Cas9.GCLC.1.AD), *Got1* (Mm.Cas9.GOT1.1.AB; Mm.Cas9.GOT1.1.AD). The following guide was used as non-targeting control: Mm.Cas9.GCGAGGTATTCGGCTCCGCG. All guides were purchased from IDT.

To prepare the targeting gRNA-Cas9 complex, equimolar amounts (180 pmol) of Alt-R CRISPR-Cas9 tracrRNA (IDT) and gene-specific crRNA (indicated above) were mixed, incubated 5 min at 98 C and cooled at room temperature for 20 min. 60 pmol of recombinant Cas9 (IDT) was mixed and incubated for 20 more min. 5–10 × $10^6$ naive CD8+ T cells isolated from OT-I mice were electroporated using the P4 Primary Cell 4D-Nucleofector™ X Kit and the prepared gRNA-Cas9 complexes. After electroporation, cells were recovered in complete medium added with components from the P4 Primary Cell 4D-Nucleofector™ X Kit and 10 ng/ml of IL-7, overnight, at 37 °C, 5% $CO_2$. Cells were then counted and further used for downstream applications.

### Glutathione quantification

For glutathione quantification, $10^5$ cells per target population were sorted and kept on ice until further processing using GSH-Glo™ Glutathione Assay (Promega), strictly following manufacturer instructions. 1 mM Tris(2-carboxyethyl)phosphin-hydrochlorid (TCEP) was used when appropriate to reduce GSSG to GSH, differentially quantify GSSG, and ultimately calculate the GSH/GSSG ratio.

### Statistical analysis

Comparisons between two groups were performed using unpaired or paired, two-tailed, Student's *t* test. Comparisons between more than two groups were performed using one-way or two-way ANOVA and Tukey's, Sidak's or Dunnet's multiple comparison test according to the experimental settings. Statistical significance is expressed throughout the text as precise *p* value. Statistical analysis was performed using Graphpad Prism 7 Software.

Statistical analysis of untargeted metabolomic data was performed applying a Student's *t* test between respectively: 1. metabolites identified in the small intestine and 2. metabolites identified in all the other tissues. This analysis was used to identify metabolites specifically present in the small intestine tissue. Analysis was performed using Metaboscape software (Bruker).

Further details on the statistical analysis can be found in the figures and the respective figure legends.

### Reporting summary

Further information on research design is available in the Nature Portfolio Reporting Summary linked to this article.

## Data availability

Data can be found under GEO accession number GSE200568, a SuperSeries containing both the scRNAseq data and bulk RNAseq data. Datasets can be found at the following link: Source data are provided with this paper.

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

## Acknowledgements

We thank members of the Department of Immunometabolism at the Max Planck Institute for Immunobiology and Epigenetics (MPI-IE) for advice and discussions throughout the project. We specially thank Katharina Borst for critical reading of the manuscript. We thank Andrea Quintana and John Sutherland for their technical support, and the Flow Cytometry, Metabolomics, Microscopy, Deep Sequencing, and Animal Facility Cores, as well as the veterinarian and animal welfare officers at the MPI-IE. We thank Federico Caicci and Francesco Boldrin of the Electron Microscopy Lab at the University of Padova for expert processing of samples. This work was supported by the Max Planck Society, the Leibniz Prize (DFG), AI156274 and AI172832 from the N.I.H. (to E.L.P.), the Fritz Thyssen Stiftung für Wissenschaftsförderung (to M.V.), an Alexander von Humboldt Postdoctoral Fellowship (to M.V) and the Medical University of Graz. R.Z. was supported by the Deutsche Forschungsgemeinschaft – SFB-1479 – Project ID: 441891347. B.B. and E.J.P. were supported by the Deutsche Forschungsgemeinschaft – IMPATH-SFB – Project ID: 256073931. M.C. was supported by the Deutsche Forschungsgemeinschaft – SFB1218 (project ID: 269925409).

## Author contributions

Study design: M.V. Conceptualization, project insight, and data interpretation: M.V., E.L.P., M.C., D.E.S. Performing and analyzing experiments: M.V., D.E.S., P.A., M.C., A.M.K., C.C., A.R., G.E.C., N.R., K.M.G., J.C., F.S., A.H., F.H., D.J.P. Providing advice and materials: M.A.S., F.B., A.M.K., A.-M.G., B.K., N.C.-W., P.H., B.B., R.Z., S., J.M.B., E.J.P. Manuscript writing: M.V., E.L.P. Funding: E.L.P., M.V.

## Competing interests

E.L.P. is a SAB member of Immunomet Therapeutics. The other authors have declared that no competing interests exist regarding this manuscript.

## Additional information

[1]Max Planck Institute for Immunobiology and Epigenetics, 79108 Freiburg, Germany. [2]Division of Rheumatology and Immunology, Department of Internal Medicine, Medical University of Graz, 8036 Graz, Austria. [3]Bloomberg-Kimmel Institute of Immunotherapy, Department of Oncology, Johns Hopkins University School of Medicine, Baltimore, MD, USA. [4]Department of Medicine I (Hematology and Oncology), University Medical Center Freiburg, 79106 Freiburg, Germany. [5]Cologne Excellence Cluster on Cellular Stress Responses in Aging-Associated Diseases (CECAD), University of Cologne, Cologne, Germany. [6]Center for Molecular Medicine (CMMC), University of Cologne, Cologne, Germany. [7]Institute for Genetics, University of Cologne, Cologne, Germany. [8]Department of Medical Biotechnology, University of Siena, Siena, Italy. [9]Department of Life Sciences, University of Trieste, 34128 Trieste, Italy. [10]Department of Medicine II, University Medical Center Freiburg, 79106 Freiburg, Germany. [11]CIBSS Centre for Integrative Biological Signalling Studies, Freiburg, Germany. [12]Faculty of Biology, University of Freiburg, 79104 Freiburg, Germany. [13]Department of Molecular Microbiology and Immunology, Bloomberg School of Public Health, Johns Hopkins University, Baltimore, MD, USA. [14]Department of Biochemistry and Molecular Biology, Bloomberg School of Public Health, Johns Hopkins University, Baltimore, MD, USA. ✉e-mail: matteo.villa@medunigraz.at; epearce6@jhmi.edu

