## [Peer Review File · Nature Communications]

Prostaglandin E2 controls the metabolic adaptation of T cells to the intestinal microenvironmentEditorial note: This manuscript has been previously reviewed at another journal that is not operating a transparent peer review scheme. This document only contains reviewer comments and rebuttal letters for versions considered at *Nature Communications*.

REVIEWER COMMENTS

Reviewer #1 (Remarks to the Author):

I am satisfied.

Reviewer #4 (Remarks to the Author):

Most of this reviewer's comments have been responded well. This reviewer acknowledges that there are technical difficulties to assess physiological/pathological impacts of PGE2 regulation of intestinal CD8 T cell metabolism in vivo.

Reviewer 3 Confidential comments to editor from last round provided with the consent of reviewer 3.

The excessive 53 page rebuttal largely presents a small number of experiments that were not incorporated into the manuscript and act as a distraction from the fact that the key concerns were not really addressed for the most part. This reviewer considers a response to their point by point a distraction from the fact the manuscript is not fundamentally changed. A few examples:

1. The addition of electron microscopy data was welcome, however, it is presented as a small number of images of portions of cells (ie no statistical/quantitative summary) and thus is of minimum impact.
2. While the authors claim to have removed the assertion that TEM give rise to TRM, this is not really case. Specific instances include in the introduction (lines 70-74) and Figure 6k the arrow from TEM to gut resident T cells. This was specifically asked to be addressed, and while the authors acknowledged this as a “misunderstanding”—the inaccurate use of these terms is damaging/confusing and is not a disagreement about nomenclature...if the same amount of effort went into revising the manuscript as did the writing of the rebuttal, then this would have been corrected.

For instance: We corrected the language in the paper to take into account the concerns of Reviewer 3. However, the finding that upon entering the gut T cells drop the mitochondrial content is clear, and this is the major point of the paper. OK, so at steady state we have no idea when the cells entered the gut without a timepoint/kinetic of the process is it the entrance into the gut or is it the cell state?

3. Further, the discussion of TRM/memory remains even after extensive defense in the rebuttal for not providing memory time points. Since the authors could not produce any memory time point data (eg response to reviewer comments highlight technical issues with later timepoints), the authors should instead compare d7 mLN vs d7 gut; memory is not assessed here. If the authors could not show the relevance of Ptger4, Atg5 and Gclc 24

days post infection, the author should report this in the published manuscript. The manuscript talks about TRM but shows no memory timepoints. If there are technical problems, they are solvable as many labs—ie the Mackay lab generate these data as recent published in NI. Alternatively, the whole paper could be reframed to focus only on effector timepoint; this would be much less interesting, however.

4. The manuscript talks about tissue adaptation. ie the last sentence of the abstract “a PGE2-autophagy-glutathione axis defines the metabolic adaption of CD8 T cells to the intestinal microenvironment”. However, the majority of the comparisons are SI to LN, this is not a tissue specific adaptation but gut vs LN. To claim this is a SI adaptation rather than tissue vs circulation. The language is important. What is the role of Ptger4/Atg5/Got1 in other tissues? This is not well explored whether at effector or memory time points, it is unclear if this is a tissue vs circulation or an adaptation to the SI.

5. Figures 1 and 2 remain entirely descriptive and superficial in spite of the suggestion to expand the analysis and ask if similar results can be gleaned/validated/expanded from analysis of existing/published datasets evaluating antigen specific T cells (ie Kurd et al datasets would be valuable for comparison as would human data, and a number of other small intestine sc or total cell RNAseq datasets that are in the literature)—are these data supported in other contexts is the question given the single timepoint, absence of infection, and insistent interpretation of a developmental trajectory in the interpretation of data?

- Many of these data suffer from multiple variables, which confound the interpretation of the data: in all of the endogenous T cell experiments, the authors are comparing cells that are both in different differentiation states and in different tissues (naive to TCM to TEM in the mLN and CD69-CD103- and CD69+CD103+ in the gut...there is not a direct relationship in TCR specificity, time since activation, etc). If only the environment is the important driver of the phenotype observed, data should be at least validated in antigen-specific, adoptively transferred T cells to eliminate the variable of time, and only focus on different/multiple tissue environments. Ie In figure 3h, the TRM population here is instead compared to naïve T cells, rather than TEM like the rest of the figure. The authors should not change their comparison group, as then it makes their conclusion difficult to interpret.

6. Since the authors could not produce any memory time point data (eg response to reviewer comments highlight technical issues with later timepoints), the authors should instead compare d7 mLN vs d7 gut. Thus, the background in the introduction covering memory cells that the study is largely addressing d7 cells, to only compare tissue environment, and memory is not assessed here. This is confusing to the reader.

7. With the variable mitotracker and TMRM data in Figure 3 (eg all cell groups show that there is not a uniform staining, that there are high and low cells, and the data for the dyes are only reported as % dye hi cells), it would be better to make less sweeping conclusions about the data, and it would be better to highlight the heterogeneity. As mitotracker green can report both mitochondrial mass and polarization, the author should, in the same experiment, have +/- verapamil efflux blocker. In supplemental 3d, the authors only treat with inhibitor, therefore, it is impossible to interpret the data. Also, There are no statistical tests shown on supplemental figure 3F, so we are to interpret that there is no statistical difference between the groups with Tom20 or Tfam western blot. Therefore, these are not confirmation of the mitotracker data, as claimed in the text (lines 183-186).

- The figure legend of supplemental figure 4B does not explain any information about the graphs. Were the number of objects normalized per cell? If not, then this could just be a

readout of how many cells were imaged.

- Supplemental data 4C & D have no quantification or statistics, therefore difficult to interpret the importance of these results.

Point-by-point response – 3rd round of revision – 13.07.2023

Nature Communications manuscript NCOMMS-23-11142A

(the original Reviewers' comments are in black, our responses are in blue)

Reviewer 1 (Remarks to the Author):

I am satisfied.

Thank you.

Reviewer 4 (Remarks to the Author):

Most of this reviewer's comments have been responded well. This reviewer acknowledges that there are technical difficulties to assess physiological/pathological impacts of PGE2 regulation of intestinal CD8 T cell metabolism in vivo.

Thank you.

Reviewer 3 Confidential comments to editor from last round provided with the consent of reviewer 3.

The excessive 53 page rebuttal largely presents a small number of experiments that were not incorporated into the manuscript and act as a distraction from the fact that the key concerns were not really addressed for the most part. This reviewer considers a response to their point by point a distraction from the fact the manuscript is not fundamentally changed. A few examples:

In our rebuttal we answered each point brought up by the Reviewers, in a comprehensive and transparent way. It is unfortunate that Reviewer 3 thinks that our work in response to the critiques has been used as a distraction. We have answered the following comments from Reviewer 3, that we were finally able to read, to attempt to further clarify their concerns.

1. The addition of electron microscopy data was welcome, however, it is presented as a small number of images of portions of cells (ie no statistical/quantitative summary) and thus is of minimum impact.

As we previously stated in our rebuttal (we refer the Editor to our extensive previous response to this point) and as we continue to stand by, electron microscopy should not be used to quantify mitochondrial content of cells. Electron microscopy lacks high throughput, and by its own nature is inadequate to quantify the extent of the mitochondrial network. Instead, the technique is used study

mitochondrial ultrastructure. We took advantage of the request of Reviewer 2 and 3 to provide additional details in our manuscript and we implemented the new findings into Extended Data Figure 4d. As we were able to isolate only a few hundred thousands of cells from the LP of the gut, we were only able to obtain a limited set of images for this tissue. The images we showed in the revised manuscript (Extended Data Figure 4d), as well as in the rebuttal letter (Reviewer Figure 18), show hints of cristae remodeling in CD8⁺ T cells isolated from the LP and the IEL fraction, supporting our statement that gut-isolated cells have dilated cristae as compared to mLN-isolated counterparts. This finding corroborates our conclusion that CD8⁺ T cells isolated from the gut have a fragmented mitochondrial network (Extended Data Figure 4b, c and d).

As requested by Reviewer 3, we have quantified the intra-cristae space in CD8⁺ T cell subsets isolated from different tissues.

Reviewer Figure 1.

Quantification of mitochondrial cristae width of T_N CD8⁺ T cells isolated from mLN, and CD69⁺ CD103⁺ CD8⁺ T cells isolated from LP and IEL of unchallenged mice, imaged by transmission electron microscopy. Quantification was performed on representative images, collected from one experiment.

2. While the authors claim to have removed the assertion that TEM give rise to TRM, this is not really case. Specific instances include in the introduction (lines 70-74) and Figure 6k the arrow from TEM to gut resident T cells. This was specifically asked to be addressed, and while the authors acknowledged this as a “misunderstanding”—the inaccurate use of these terms is damaging/confusing and is not a disagreement about nomenclature...if the same amount of effort went into revising the manuscript as did the writing of the rebuttal, then this would have been corrected.

We have now amended the sentence in the introduction:

From: “There, after clearing the antigen that initiated the immune response, only a small proportion of T_{EM} survive; some continue to circulate in and out of peripheral tissues, while others acquire residency in the intestine. Moreover, during the early phases of the immune response, a fraction of $CD8^+$ T cells commits to the tissue-resident memory T cell (T_{RM}) fate, eventually seeding the gut tissue.”

To: “There, after clearing the antigen that initiated the immune response, only a small proportion of T_{EM} survives, and continues to circulate in and out of peripheral tissues. Moreover, during the early phases of the immune response, a fraction of $CD8^+$ T cells commits to the tissue-resident memory T cell (T_{RM}) fate, eventually seeding the gut tissue.”

Also, in Figure 6k, the arrow from T_{EM} to T_{RM} has been removed.

For instance: We corrected the language in the paper to take into account the concerns of Reviewer 3. However, the finding that upon entering the gut T cells drop the mitochondrial content is clear, and this is the major point of the paper. OK, so at steady state we have no idea when the cells entered the gut without a timepoint/kinetic of the process is it the entrance into the gut or is it the cell state?

We think that what defines the reduction of mitochondrial content in gut-isolated $CD8^+$ T cells is the entrance in the gut and the cells' ability to sense the gut microenvironment. PGE_2 is an important part of it, and its sensing plays a role in regulating the mitochondrial content of gut-isolated cells. In the mouse small intestine, at the steady state, $CD69^+ CD103^+ CD8^+$ T cells are defined as tissue-resident memory cells. However, we found that also $CD8\alpha\alpha$ T cells in the intraepithelial lymphocyte fraction, thus not classically defined as $CD8^+ T_{RM}$ cells, show low mitochondrial content (Reviewer Figure 1). We do not think that the cell state itself (in this case the T_{RM} state) drives the reduction of the mitochondrial content, as T_{RM} cells isolated from the mLN have a comparable mitochondrial content to other $CD8^+$ T cells subsets resident in the mLN (Figure 3b).

As correctly pointed out by Reviewer 3, we cannot state what the kinetic of the mitochondrial content reduction is during the transition from mLN to the gut wall. However, as shown in Extended Data Figure 4a, antigen-specific $CD8^+$ T cells isolated 24 days after oral delivery of *LmOVA*, showed a gut-specific reduction of the mitochondrial content, at time points compatible with T_{RM} formation. We think these data support our conclusion that localization into the gut affects the mitochondrial content of $CD8^+$ T cells.

In other words, we think that the *spatial* location of $CD8^+$ T cells in the gut as compared to mLN, rather than their differentiation kinetics, shape the mitochondrial content of the cells.

We have now amended the whole manuscript to avoid any statement that may hint to the requirement of the reduction of mitochondrial content for the acquisition of the T_{RM} fate, or any kinetic implication of our statements (marked in green in the new version of the manuscript).

We hope we have correctly understood this comment of Reviewer 3 and properly addressed it.

3. Further, the discussion of TRM/memory remains even after extensive defense in the rebuttal for not providing memory time points. Since the authors could not produce any memory time point data (eg response to reviewer comments highlight technical issues with later timepoints), the authors should instead compare d7 mLN vs d7 gut; memory is not assessed here. If the authors could not show the relevance of *Ptger4*, *Atg5* and *Gclc* 24 days post infection, the author should report this in the published manuscript. The manuscript talks about TRM but shows no memory timepoints. If there are technical problems, they are solvable as many labs—ie the Mackay lab generate these data as recent published in NI. Alternatively, the whole paper could be reframed to focus only on effector timepoint; this would be much less interesting, however.

We disagree with Reviewer 3 that our data would be much less interesting if they would only refer to the effector timepoints.

We have toned down memory-related statements. We also added a sentence in the discussion (marked in purple) to state that the role of *Ptger4*, *Atg5* and *Gclc* are yet to be investigated at memory time points.

Finally, we want to point out that many of our data have been obtained from analysis of the intestine at steady-state, where tissue-isolated CD8⁺ T cells are characterized by the co-expression of CD69 and CD103 and defined as tissue-resident memory T cells. Further, data shown in Extended Data Figure 4a refer to antigen-specific CD8⁺ T cells analysed 24 days after *LmOVA* challenge.

4. The manuscript talks about tissue adaptation. ie the last sentence of the abstract “a PGE2-autophagy-glutathione axis defines the metabolic adaption of CD8 T cells to the intestinal microenvironment”. However, the majority of the comparisons are SI to LN, this is not a tissue specific adaptation but gut vs LN. To claim this is a SI adaptation rather than tissue vs circulation. The language is important. What is the role of *Ptger4/Atg5/Got1* in other tissues? This is not well explored whether at effector or memory time points, it is unclear if this is a tissue vs circulation or an adaptation to the SI.

We do not understand this comment of Reviewer 3.

We compare CD8⁺ T cells isolated from the mLN (where naïve CD8⁺ T cells are localized and represent the “starting point” of the CD8⁺ T cell-driven intestinal immune response) and from the LP and IEL of the intestine. We think it is fair to claim that during the transition from mLN to LP and ultimately to IEL, CD8⁺ T cells adapt to the intestinal microenvironment.

As shown in Figure 4a, we do not think that reduction of mitochondrial content is an adaptation occurring in every tissue to the same extent. For comparison to other tissues, we refer Reviewer 3 to the data shown in Reviewer Figure 7 (comparison of mitochondrial content between mLN, gut and female reproductive tract), Reviewer Figure 8 (expression of autophagy and glutathione genes between first mLN and liver), Reviewer Figure 9 (day 7 distribution of CD8⁺ T cells in the liver upon co-transfer of *Atg5*-deficient and *Gclc*-deficient cells), and Reviewer Figure 36 (day 7 distribution of CD8⁺ T cells in blood and liver upon co-transfer of *Ptger4*-deficient, *Atg5*-deficient and *Gclc*-deficient cells). We left these data to the Reviewers' consideration only as our manuscript aims at dissecting the adaptation of CD8⁺ T cells to the intestinal environment and it does not aim to make claims referring to other tissues.

5. Figures 1 and 2 remain entirely descriptive and superficial in spite of the suggestion to expand the analysis and ask if similar results can be gleaned/validated/expanded from analysis of existing/published datasets evaluating antigen specific T cells (ie Kurd et al datasets would be valuable for comparison as would human data, and a number of other small intestine sc or total cell RNAseq datasets that are in the literature)—are these data supported in other contexts is the question given the single timepoint, absence of infection, and insistent interpretation of a developmental trajectory in the interpretation of data?

We believe that single cell RNA sequencing datasets are prone to originate descriptive data. The originality of our approach is that we analysed at the same time CD8⁺ T cells isolated from mLN, LP and IEL. LP is often not included in these analysis efforts, perhaps because *bona fine* CD8⁺ T_{RM} cells can be defined in the IEL fraction, whereas cells in the LP may represent an early stage of their differentiation pathway. However, for the purpose of our study and since we are not focusing on T_{RM}, the LP represents the key tissue compartment to analyse as it is the entry point of cells in the gut wall.

As already mentioned in our previous answer to Reviewer 3 (points 2b and 2c), our single cell RNA sequencing analysis provided key and novel insights into the differential expression of mitochondrial genes, and it was designed as a *starting point* to explore the adaptation of CD8⁺ T cells to the intestinal microenvironment, to be followed up with a thorough analysis of the mitochondrial phenotype of CD8⁺ T cells isolated from the mLN and the gut wall, that we performed later on in the manuscript.

It is fair to point out that during the previous round of Reviewer's 3 comments there was no request of performing additional analysis of our single cell RNA sequencing dataset, nor of previously published ones (see points 2b and 2c, where the criticisms of Reviewer 3 were addressed). If editorially required, we could re-analyze previously published datasets as mentioned now by Reviewer 3, however, we think that this request is unjustified, considering the fact that our analysis

of mitochondrial content in CD8⁺ T cells (Figure 3 and Extended Data Figure 4), including in the antigen-specific setting (Figure 3), confirmed our transcriptional data.

Finally, as pointed out in the previous version of the answer to Reviewer 3, we do not aim to interpret nor make any claim regarding the developmental trajectory of CD8⁺ T cells. This is the reason why, after the previous comments of Reviewer 3, the Slingshot analysis was moved into the Extended Data section. We also want to reiterate that we do not use the Slingshot analysis in the direction of interpreting the developmental trajectories of CD8⁺ T cells, but rather to explore the transcriptional adaptation of CD8⁺ T cells to different spatial locations.

- Many of these data suffer from multiple variables, which confound the interpretation of the data: in all of the endogenous T cell experiments, the authors are comparing cells that are both in different differentiation states and in different tissues (naïve to TCM to TEM in the mLN and CD69-CD103- and CD69+CD103+ in the gut...there is not a direct relationship in TCR specificity, time since activation, etc). If only the environment is the important driver of the phenotype observed, data should be at least validated in antigen-specific, adoptively transferred T cells to eliminate the variable of time, and only focus on different/multiple tissue environments. In Figure 3h, the TRM population here is instead compared to naïve T cells, rather than TEM like the rest of the figure. The authors should not change their comparison group, as then it makes their conclusion difficult to interpret.

We used well accepted surface markers to identify the CD8⁺ T cell subsets that we analysed for their mitochondrial content. In terms of the validation of our phenotype in an antigen-specific setting, we refer Reviewer 3 to Figures 3d and 3f (d7 post *LmOVA* challenge) as well as to Extended Data Figure 4a (day 24 post *LmOVA* challenge). In Figure 3h, we did not compare CD69⁺CD103⁺ cells in the LP with mLN T_{EM} because, due to cell number constraints, the metabolomic analysis was performed on a mixed population of T_{EM} and T_{RM} cells, as indicated. If required, we can remove the statistical analysis in Figure 3h and simply leave the single data distribution.

6. Since the authors could not produce any memory time point data (eg response to reviewer comments highlight technical issues with later timepoints), the authors should instead compare d7 mLN vs d7 gut. Thus, the background in the introduction covering memory cells that the study is largely addressing d7 cells, to only compare tissue environment, and memory is not assessed here. This is confusing to the reader.

After the latest amendments of the manuscript, and the answers to the previous points raised by Reviewer 3, we believe this concern has been addressed.

7. With the variable mitotracker and TMRM data in Figure 3 (eg all cell groups show that there is not a uniform staining, that there are high and low cells, and the data for the dyes are only reported as % dye hi cells), it would be better to make less sweeping conclusions about the data, and it would be better to highlight the heterogeneity. As mitotracker green can report both mitochondrial mass and polarization, the author should, in the same experiment, have +/- verapamil efflux blocker. In supplemental 3d, the authors only treat with inhibitor, therefore, it is impossible to interpret the data. Also, There are no statistical tests shown on supplemental figure 3F, so we are to interpret that there is no statistical difference between the groups with Tom20 or Tfam western blot. Therefore, these are not conformation of the mitotracker data, as claimed in the text (lines 183-186). - The figure legend of supplemental figure 4B does not explain any information about the graphs. Were the number of objects normalized per cell? If not, then this could just be a readout of how many cells were imaged.

As stated in our previous answer to the Reviewers' comments, our Mitotracker data have been validated upon using of multiple controls (Extended Data Figures 3d and e), and with different methods (Figure 3c, Extended Data Figure 4b). We agree that the Mitotracker Green staining profile shown in Figure 3b is heterogeneous, and this is the reason why we use the parameter % of Mitotracker^{high} cells. If we would have used the geometric MFI, it would not have correctly represented the profile of our cell populations. We would also like to point out that upon verapamil treatment, the Mitotracker profile acquires a bell-shaped distribution, hence our use of geometrical MFI.

We used Verapamil to prevent the efflux of Mitotracker Green from mitochondria mediated by the proteins Bcrp1 and Mdr1a/b, not to distinguish between mitochondrial content and mitochondrial membrane potential. This misunderstanding may rise from our previous answer that was perhaps unclear (see our previous rebuttal to point 1 of Reviewer 3). As we pointed out in Extended Data Figure 3d, the meaningful comparison is between the Mitotracker Green geometric MFI of CD8⁺ T cells isolated from the mLN and the counterparts isolated from the intestine. We are not aiming at evaluating a possible differential susceptibility of CD8⁺ T cells to Verapamil. To satisfy the request of Reviewer 3, we performed the requested experiment and we are showing the Mitotracker Green results with or without Verapamil in the graph below (**Reviewer Figure 2**):

Reviewer Figure 2.

Flow cytometry analysis of Mitotracker Green staining with or without Verapamil in the indicated cell subsets isolated from mLN, and small intestine LP and IEL fraction of steady-state mice. Data of n = 1 experiment.

Following Reviewer 3 suggestion, we performed statistical analysis of Extended Data Figure 3f, comparing mLN T_{EM} and LP CD69⁺CD103⁺ cells. The *p*-value for the Tom20 blot is equal to 0.0484, whereas the *p*-value for the TFAM blot is equal to 0.0939. We toned down the text in the manuscript accordingly (see text marked in orange).

Regarding Extended Data Figure 4b, each dot represents the number of objects (mitochondria) per cell. The number of dots thus represent the number of cells quantified. We amended the figure legend accordingly (see text marked in light blue).

- Supplemental data 4C & D have no quantification or statistics, therefore difficult to interpret the importance of these results.

The western blot in Extended Data Figure 4c is qualitative in nature, as we compare the distribution of low vs high molecular weight isoforms of Opa-1. If editorially required, we will calculate the ratio.

REVIEWER COMMENTS

Reviewer #5 (Remarks to the Author):

The manuscript by Villa et.al. addresses metabolic characterizations of CD8+ T cells across mLN, LP, and IEL tissues. Much of the prior review focused on the concern that some of the data centers on the analysis of polyclonal CD8 T cells in the steady state which makes it impossible to fully know the differentiation state or history of a cell and the timing of entry into the tissue. Be that as it may, the cells when isolated from these respective locations present a clear difference in mitochondrial mass that is linked to PGE2 sensing and seems unique to T cells in the LP and IEL. Importantly, appropriate definitions and gating schemes for each T cell population are included. In addition, some experiments using OT-1 adoptive transfer followed by oral LM-OVA infection have been added to bolster conclusions.

Overall, I think reasonable experimental strategies have been used and the authors have sufficiently addressed the concerns of prior review. The language edits and data additions have appropriately balanced conclusions and added necessary details. I have one minor comment which relates to clarity, but I am supportive of publication.

1. Figure 3d,f- CD90.1+ cells should be labeled as OT-1 as they are in later Figures in the paper.

Reviewer #6 (Remarks to the Author):

Nature Communications manuscript NCOMMS-23-11142A

Prostaglandin E2 controls the metabolic adaptation of T cells to the intestinal microenvironment

A) My comments and thoughts on the first round of review by reviewer nr 3

I totally agree with reviewer 3 that figures 1 and 2 (and in my opinion also figure 3) are truly superficial and excessive in this paper. Several published studies are available that provide much more in-depth analyses, including single cell analysis of T cells transitioning from the naïve stage to TRM, of cells transitioning to the intestine, as well as data showing the phenomena that T cells that migrate and reside in the intestine undergo adjustments to lower down their metabolic activity including mitochondrial content. The data here are of a lower quality or lesser detail.

I disagree with reviewer 3 on the importance of recapitulating the precise process of naïve to TE to TEMP/TCM/TEM to TRM in the context of this paper. I agree with the authors, it's not about the adaptation to TRM cells but to the gut (epithelial) microenvironment.

B) My comments and thoughts on the second round of review by reviewer nr 3/ author's rebuttal.

1) I disagree with reviewer 3 on the additional electron microscope data comment. The new data further support the initial finding here and already published finding, stating that the mitochondrial content of the T cells that accumulate and reside long-time in the gut

epithelium, declines over time.

- Comparison of mucosal T cells and different T cell types/stages (my expertise)

I agree with the authors, that this study is not about T cell memory or memory formation at different stages, but rather focused on the specific transition T cells undergo as they adapt and reside long-term in the gut (epithelium).

2) I also strongly agree with Reviewer 3 that the comparisons made in this paper are not appropriate and contain too many variables (the apples with pears comparison is very obvious in this paper)

a) I believe that the authors do a poor job on describing the different intestinal T cells.

From the description in this paper, it looks like LP cells and epithelial T cells are one and the same, when in fact they are extremely different and reside in very different microenvironments. This is not at all appropriately addressed here. The cells are basically treated as one and the same. Also, the isolation and purification of LP and epithelial T cells is a concern. Judging from the data, LP cells have a lot of CD103+ cells, which is unusual and might be an indication that they have contaminating intraepithelial T cells in their LP preps.

Importantly, the adjustment to the microenvironment they describe here is typical for the epithelial T cells specifically (marked by the expression of CD103).

Sometimes the authors use gut wall cells to indicate T cells in the LP and epithelium and at other times they use gut wall to distinguish epithelium from the LP. This cannot be used at random, since the two environments, cell types and adjustments made are very different. This should be properly addressed, especially in this kind of study.

b) It is also not possible to compare pathogen-induced effector T cells with those generated at steady state under quiescent conditions. They do not follow the same trajectory at all. Priming of naïve CD8ab T cells at steady state (mostly by diet Ags under quiescent conditions) does not necessarily occur in the mLN and does not lead to TRM. Instead CD8ab T cells generated at steady state starting during weaning, require constant Ag-exposure for their maintenance in the epithelium. Therefore, they are not typical TRM. So, CD8ab T cells in the gut generated at steady state and upon OVA-Lm infection, are not comparable.

c) Even if pathogen-induced T cells are looked at, the authors should look and compare pathogen-responding (OT-I) CD8ab T cells in the gut induced upon an oral infection with Lm-OVA compared to a nasal infection for example. Although, the OT-I CD8ab T cells and the pathogen/Antigen (Lm-OVA) are the same, the different route of infection does or does not generate TRM cells in the gut (Sheridan et al., 2014 PMID: 24792910).

In this case the naïve T cells are identical: OT-I, the antigen/infection is the same: LmOVA, and the gut environment is the same. In both cases CD8ab OT-I T cells accumulate in the gut but in the oral infection case they become TRM, in the nasal infection they don't. Is PGE2-driven mitochondrial modulation involved in both? (that would mean gut tissue adaptation) or just with the oral infection, which generates TRM?

These would have been appropriate comparisons to address the issue and to have the appropriate comparable and variables.

d) The steady state CD8ab mucosal T cells, which in contrast to the pathogen-induced CD8ab T cells, are generated under quiescent conditions (or non-inflammatory conditions) should have been compared with other mucosal T cell subsets that are present at steady state. This is very important, especially in terms of the PGE2-driven mechanism that is highlighted here. It is almost certain that the PGE2 release (dose) and (source) will be different under quiescent versus inflammatory conditions and given that PGE2 effects are so dose dependent (shown in many published studies), this should be properly looked at with a comparison between different T cell types in the epithelium (CD8ab, CD4CD8aa+, and TCRab and TCRgd CD8aa T cells in the epithelium and CD4 Th17/Th1 and Treg and CD8ab T cells in the LP). If all of these cell types, which have different origins and specificities but are all in the same gut microenvironment and present at steady state, also all display the downsizing in metabolic activity and mitochondrial content, then these data would indeed strongly support the phenomena as an adaptation to the gut microenvironment and not an activation stage or functional fate of the T cell.

e) Finally, authors could have compared naïve OT-I T cells transferred to RAG-deficient recipient mice, which will migrate to the intestine as naïve cells but will not transition to effector cells or TRM cells, whereas feeding the recipients an OVA-diet will induce functional maturation of the OT-I cells in the tissue. Comparison of these two OT-I donor cell populations in the gut in terms of their metabolism/mitochondrial adaptation, would have been telling about the mechanisms of gut adaptation, with the same cells and the same environment but different Ag conditioning.

Comments on the metabolic aspects/data. Although not directly my expertise, I do have some comments and concerns

f) The finding that PGE2 participates in the metabolic regulation of the T cells as part of the adaptation to the gut environment is interesting and novel. However, the study is rather superficial. It is not clear how PGE2 is controlling the metabolic state of the T cells. It is also not clear why PGE2 would specifically act on the mucosal T cells.

PGE2 effects are dose-dependent and different doses can have opposite effects.

Who is providing the PGE2? The activated T cells? The IECs?

The PGE2 release has to be local since PGE2 is unstable.

Is PGE2 also controlling the reversal state of reactivation in case of a secondary challenge?

What controls that switch from inactive to active? Is it also driven by metabolic events?

g) I agree with reviewer #3, that the in vitro system with IL15/TGFb is not appropriately reflecting the in vivo situation in the gut. Moreover, PGE2 has significant effects on TGFb signaling. Are both needed in vivo for the effect on the T cell metabolism in the gut? If so it could explain, why PGE2 has such specific effect in the gut, where TGFb is abundantly expressed and central for controlling several gut adaptation processes.

Unfortunately, this was not addressed or even considered in this paper. Overall, although the PGE2 aspect is new and interesting, it is poorly developed and in agreement with reviewer

3, this study leaves the reader with many more questions than answers.

Point-by-point response – 4th round of revision – 10.10.2023

Nature Communications manuscript NCOMMS-23-11142A

Reviewer #5 (Remarks to the Author):

The manuscript by Villa et.al. addresses metabolic characterizations of CD8+ T cells across mLN, LP, and IEL tissues. Much of the prior review focused on the concern that some of the data centers on the analysis of polyclonal CD8 T cells in the steady state which makes it impossible to fully know the differentiation state or history of a cell and the timing of entry into the tissue. Be that as it may, the cells when isolated from these respective locations present a clear difference in mitochondrial mass that is linked to PGE2 sensing and seems unique to T cells in the LP and IEL. Importantly, appropriate definitions and gating schemes for each T cell population are included. In addition, some experiments using OT-1 adoptive transfer followed by oral LM-OVA infection have been added to bolster conclusions.

Overall, I think reasonable experimental strategies have been used and the authors have sufficiently addressed the concerns of prior review. The language edits and data additions have appropriately balanced conclusions and added necessary details. I have one minor comment which relates to clarity, but I am supportive of publication.

1. Figure 3d,f- CD90.1+ cells should be labeled as OT-1 as they are in later Figures in the paper.

We have now amended the legends in figures 3d and 3f (now figures 2d and 2f), and in the corresponding text of the figure legends (see amendments in green).

Reviewer #6 (Remarks to the Author):

Amendments to address the remarks of Reviewer 6 are highlighted in red in the new version of the manuscript.

Nature Communications manuscript NCOMMS-23-11142A

Prostaglandin E2 controls the metabolic adaptation of T cells to the intestinal microenvironment

A) My comments and thoughts on the first round of review by reviewer nr 3

I totally agree with reviewer 3 that figures 1 and 2 (and in my opinion also figure 3) are truly superficial and excessive in this paper. Several published studies are available that provide much more in-depth analyses, including single cell analysis of T cells transitioning from the naïve stage to TRM, of cells transitioning to the intestine, as well as data showing the phenomena that T cells that migrate and reside in the intestine undergo adjustments to lower down their metabolic activity including mitochondrial content. The data here are of a lower quality or lesser detail.

Following the hint of Reviewer 6, we now merged figures 1 and 2 to streamline the manuscript, while moving some data to supplementary figures. We left figure 3 (now figure 2) unaltered as it is essential for the structure of the manuscript. This figure shows novel, comprehensive, and functional data regarding mitochondria metabolism in CD8⁺ T cells isolated from the intestinal tissue, spanning both mice and humans.

I disagree with reviewer 3 on the importance of recapitulating the precise process of naïve to TE to TEMP/TCM/TEM to TRM in the context of this paper. I agree with the authors, it's not about the adaptation to TRM cells but to the gut (epithelial) microenvironment.

B) My comments and thoughts on the second round of review by reviewer nr 3/ author's rebuttal.

1) I disagree with reviewer 3 on the additional electron microscope data comment. The new data further support the initial finding here and already published finding, stating that the mitochondrial content of the T cells that accumulate and reside long-time in the gut epithelium, declines over time.

- Comparison of mucosal T cells and different T cell types/stages (my expertise)

I agree with the authors, that this study is not about T cell memory or memory formation at different stages, but rather focused on the specific transition T cells undergo as they adapt and reside long-term in the gut (epithelium).

2) I also strongly agree with Reviewer 3 that the comparisons made in this paper are not appropriate and contain too many variables (the apples with pears comparison is very obvious in this paper)

a) I believe that the authors do a poor job on describing the different intestinal T cells.

From the description in this paper, it looks like LP cells and epithelial T cells are one and the same, when in fact they are extremely different and reside in very different microenvironments. This is not at all appropriately addressed here. The cells are basically treated as one and the same.. Also, the isolation and purification of LP and epithelial T cells is a concern. Judging from the data, LP cells have a lot of CD103⁺ cells, which is unusual and might be an indication that they have contaminating intraepithelial T cells in their LP preps.

Importantly, the adjustment to the microenvironment they describe here is typical for the epithelial T cells specifically (marked by the expression of CD103).

Sometimes the authors use gut wall cells to indicate T cells in the LP and epithelium and at other times they use gut wall to distinguish epithelium from the LP. This cannot be used at random, since the two environments, cell types and adjustments made are very different. This should be properly addressed, especially in this kind of study.

We agree with the comment of Reviewer 6 stating that the LP and IEL compartments are very different. This is the reason why in our figures LP-isolated and IEL-isolated CD8⁺ T cells have been shown independently (throughout the manuscript, CD69⁺CD103⁺ cells from the LP have been distinguished from CD69⁺CD103⁺ cells from the IEL fraction). Following the suggestion of Reviewer 6, we amended the manuscript and rephrased the text so to consistently state this separation when mentioning the results related to cells isolated from LP or IEL. Also, in the amended version of the manuscript, we limited the use of the expression “gut wall” and rather specified whether the cells were originating from the LP or the IEL fraction.

The presence of CD103⁺ CD8⁺ T cells in the LP is not unusual. We refer Reviewer 6 to the recently published paper from Fung EY, Sci Immunol 2022 (PMID: 36332012), specifically to figure 1B that shows that both LP and IEL compartments of mice at steady-state are characterized by the presence of CD103⁺ cells within the fraction of CD8⁺ CD69⁺ T cells. Moreover, we are confident in saying that our LP-isolated cells are not contaminated by intraepithelial T cells for the following reasons.

1. In our hands, the IEL fraction contains a substantial proportion of CD8 $\alpha\alpha$ ⁺ T cells (around 60%), known to be abundant within the epithelial layer of mucosal tissues. On the other hand, suspensions isolated from the LP contain a very limited proportion of CD8 $\alpha\alpha$ ⁺ T cells (around 2%). If intraepithelial T cells would be a substantial contaminant of the LP fraction, we would expect the LP fraction to show a higher representation of CD8 $\alpha\alpha$ ⁺ T cells, to closely resemble the cellular composition of IEL (**Reviewer 6 – figure 1**).

Reviewer 6 – figure 1

Flow cytometry analysis of cellular composition of suspensions isolated from LP and IEL of naïve C57BL/6J mice. Lines in the dot plots show mean values. Data are representative of n = 9 experiments.

- CD8 $\alpha\beta$ ⁺ T cells isolated from LP and IEL have undergone RNA sequencing. As shown in supplementary figure 4e (bulk RNA sequencing), LP-isolated CD69⁺CD103⁺ cells are transcriptionally different from CD69⁺CD103⁺ T cells isolated from the IEL fraction, as there is no overlap in the distribution of the two cell populations. Also, when considering the single cell RNA sequencing data (figure 1c and 1d), CD8 T cells isolated from the LP and the IEL are segregated in different, non-overlapping clusters, and occupy different areas of the UMAP space. Contamination of IEL cells in LP suspensions would result in overlapping transcriptional profiles.
- While in terms of mitochondrial content, CD69⁺CD103⁺ cells from the LP resemble CD69⁺CD103⁺ cells from the IEL, these cell subsets also have distinctive metabolic features. As compared to cells isolated from the IEL, cells isolated from the LP show much higher phosphorylation of the ribosomal protein S6, a proxy of mTORC1 activation, positively correlated to activation of glycolysis (**Reviewer 6 – figure 2**). Such a divergent result does not support the possibility that a substantial contamination of IEL-derived cells is found in the LP suspensions.

Reviewer 6 – figure 2

Intracellular flow cytometry analysis of phosphorylation of S6 in the indicated cell populations isolated from mLN, LP and IEL of naïve C57BL/6J mice. Positive control using PMA stimulation to assess technical success of the

intracellular staining has been performed, but not shown. Lines in the dot plots show mean values. Data are representative of n = 2 experiments.

b) It is also not possible to compare pathogen-induced effector T cells with those generated at steady state under quiescent conditions. They do not follow the same trajectory at all. Priming of naïve CD8ab T cells at steady state (mostly by diet Ags under quiescent conditions) does not necessarily occur in the mLN and does not lead to TRM. Instead CD8ab T cells generated at steady state starting during weaning, require constant Ag-exposure for their maintenance in the epithelium. Therefore, they are not typical TRM. So, CD8ab T cells in the gut generated at steady state and upon OVA-Lm infection, are not comparable.

We agree with Reviewer 6 regarding the comparison between steady-state polyclonal T cells and OVA-specific effector cells. However, it was never our intention to directly compare those two cell types, as the antigen-specificity is different, as well as the developmental kinetic of those cells are likely to differ, as pointed out by Reviewer 6. Within polyclonal steady-state CD8⁺ T cells and OVA-specific CD8⁺ T cells, we always compared cells isolated from the mLN and cells isolated from the intestinal compartments of LP and IEL. The reason why we used OT-I cells was to confirm our findings in an antigen-specific setting, as well as to set up a system that allowed CRISPR-Cas9 gene editing to test our hypotheses.

We thank Reviewer 6 for pointing out the atypical nature of CD69⁺CD103⁺ cells isolated from the LP and IEL of unchallenged mice. Following up on this comment, we amended the text and now refrain from defining IEL-isolated CD69⁺CD103⁺ cells as T_{RM}, in line with what we have done for LP-isolated cells. Also, we modified figure 2d and 2f, clearly separating the comparisons performed within polyclonal CD90.1⁻ T cells, and the ones done within antigen-specific CD90.1⁺ OT-I cells.

c) Even if pathogen-induced T cells are looked at, the authors should look and compare pathogen-responding (OT-I) CD8ab T cells in the gut induced upon an oral infection with Lm-OVA compared to a nasal infection for example. Although, the OT-I CD8ab T cells and the pathogen/Antigen (Lm-OVA) are the same, the different route of infection does or does not generate TRM cells in the gut (Sheridan et al., 2014 PMID: 24792910).

In this case the naïve T cells are identical: OT-I, the antigen/infection is the same: LmOVA, and the gut environment is the same. In both cases CD8ab OT-I T cells accumulate in the gut but in the oral infection case they become TRM, in the nasal infection they don't. Is PGE2-driven mitochondrial modulation involved in both? (that would mean gut tissue adaptation) or just with the oral infection, which generates TRM?

These would have been appropriate comparisons to address the issue and to have the appropriate comparable and variables.

We thank Reviewer 6 for clearly explaining how to address the previous comment of Reviewer 3. We indeed did not understand that comment of Reviewer 3.

As we stated above, and in the previous rounds of answers to Reviewer 3, our paper does not aim to link the reduction of mitochondrial content observed in gut-isolated cells to the acquisition of the T_{RM} fate (as also Reviewer 6 recognized at the beginning of his/her comments). Because of this reason, and because we are logistically unable to perform the suggested experiments (move of the lab to another country and having to write new animal protocols to perform the experimental work required), we cannot satisfy this request of Reviewer 6.

However, we provide additional experimental data that may address this comment of Reviewer 6. Recipient mice transferred with naïve OT-I cells were orally infected with *LmOVA*. At day 7 and day 24 post-infection, we isolated CD8⁺ T cells from the LP of the gut, and using the surface markers CD69 and CD103, we assessed whether both CD69⁻CD103⁻ and CD69⁺CD103⁺ cells isolated from the LP reduced the mitochondrial content as compared to OT-I cells isolated from the mLN. We found that both the populations reduced their mitochondrial content as compared to OT-I cells isolated from the mLN, suggesting that the intestinal microenvironment, rather than the acquisition of the CD69⁺CD103⁺ phenotype (T_{RM}, at least at day 24 post *LmOVA* infection), shapes the metabolic profile of CD8⁺ T cells (**Reviewer 6 – figure 3**).

Reviewer 6 – figure 3

Flow cytometry analysis of Mitotracker Green staining in the indicated OT-I cell populations isolated from mLN, and small intestine LP and IEL fraction of *LmOVA* infected mice, performed 7 and 24 days post-infection. Lines in the dot plots show mean values. Representative data of n = 4 experiments (day 7) and n = 2 experiments (day 24). Statistics were performed using one-way ANOVA and Tukey's multiple comparison correction.

We then assessed whether *Ptger4* deletion impaired the reduction of mitochondrial content in CD69⁻CD103⁻ cells isolated from the LP, as we observed in CD69⁺CD103⁺ cells (Figure 3j). In this case, *Ptger4* deletion did not uniformly prevent the loss of mitochondrial content in CD69⁻CD103⁻ cells isolated from the LP, suggesting that other additional mechanisms may play a role in the regulation of mitochondrial content upon entry of T cells in the gut microenvironment, as we also suggested in the discussion of our manuscript (**Reviewer 6 – figure 4**).

Reviewer 6 – figure 4

Flow cytometry analysis of Mitotracker Green staining in Ctrl vs *Ptger4*-deficient CD69⁻CD103⁻ OT-I cells isolated from the LP of *LmOVA* infected mice, 7 days post-infection. Lines in the plot connect paired samples. Cumulative data of n = 3 experiments. Statistics were performed using a two-tailed paired Student's *t* test.

Finally, to further reiterate that the reduction of mitochondrial content observed in CD8⁺ T cells isolated from the LP and IEL compartments of the small intestine is not linked to the acquisition of a T_{RM} phenotype, we point Reviewer 6 to the data shown in Figure 3a. CD8⁺ T cells isolated from liver, lung and mLN, labelled by the markers CD69 and/or CD103, do not reduce their mitochondrial content to the extent of cells isolated from the gut, suggesting that the acquisition of the CD69⁺CD103⁺ phenotype *per se* does not correlate with reduction of the mitochondrial content.

Following the suggestion of Reviewer 1, and to now back up our answer to Reviewer 6, we analysed the female reproductive tract (FRT) known to harbour CD8⁺ T cells. We found that unlike CD8⁺ T cells isolated from the LP and IEL of the small intestine, CD8⁺ T cells isolated from the FRT do not reduce their mitochondrial content, suggesting that the gut microenvironment rather than tissue localization drives of this metabolic adaptation (**Reviewer 6 – figure 5**).

Reviewer 6 – figure 5

Flow cytometry analysis of Mitotracker Green staining in CD8⁺ T cell subsets isolated from the mLN, LP and female reproductive tract (FRT) of unchallenged C57BL/6J mice. Lines in the dot plots show mean values. Data are representative of n = 2 independent experiments.

d) The steady state CD8ab mucosal T cells, which in contrast to the pathogen-induced CD8ab T cells, are generated under quiescent conditions (or non-inflammatory conditions) should have been compared with other mucosal T cell subsets that are present at steady state. This is very important, especially in terms of the PGE2-driven mechanism that is highlighted here. It is almost certain that the PGE2 release (dose) and (source) will be different under quiescent versus inflammatory conditions and given that PGE2 effects are so dose dependent (shown in many published studies), this should be properly looked at with a comparison between different T cell types in the epithelium (CD8ab, CD4CD8aa+, and TCRab and TCRgd CD8aa T cells in the epithelium and CD4 Th17/Th1 and Treg and CD8ab T cells in the LP). If all of these cell types, which have different origins and specificities but are all in the same gut microenvironment and present at steady state, also all display the downsizing in metabolic activity and mitochondrial content, then these data would indeed strongly support the phenomena as an adaptation to the gut microenvironment and not an activation stage or functional fate of the T cell.

Following the suggestions of Reviewers 1, 2 and now 6 we assessed the mitochondrial content in mucosal-associated cells other than CD8 $\alpha\beta$ ⁺ T cells.

We assessed Mitotracker Green staining in CD8 $\alpha\alpha$ ⁺ cells isolated from the IEL fraction of the small intestine. As observed for CD8 $\alpha\beta$ ⁺ cells, CD8 $\alpha\alpha$ ⁺ cells showed a remarkably reduced amount of mitochondria, as compared to cells isolated from the mLN (**Reviewer 6 – figure 6**).

Reviewer 6 – figure 6

Flow cytometry analysis of Mitotracker Green staining in cell subsets isolated from the mLN, LP and IEL of unchallenged C57BL/6J mice. Color coding: Grey: mLN CD44⁺CD62L⁻CD69⁻CD103⁻ T cells; Green: LP CD69⁻CD103⁻ T cells; Orange: LP CD69⁺CD103⁺ T cells; Blue: IEL CD8 $\alpha\beta$ ⁺; Red: IEL CD8 $\alpha\alpha$ ⁺ cells. Data are representative of n = 9 independent experiments.

We also tested whether the same applied to T_H17 cells. Using *IL17a^{Cre} x R26R^{eYFP}* mice, we assessed

the staining of Mitotracker Deep Red in T_H17 cells isolated from the mLN and LP of the small intestine. Similarly to what observed for other T cell subsets isolated from the LP, and IEL, LP T_H17 have a substantially lower staining of Mitotracker Deep Red as compared to counterparts isolated from the mLN (**Reviewer 6 – figure 7**). Of note, Mitotracker Deep Red measures the mitochondrial transmembrane potential, similarly to TMRM. We could not use Mitotracker Green due to the fluorescence interference with eYFP.

Reviewer 6 – figure 7

Flow cytometry analysis of Mitotracker Deep Red staining in T_H17 cells isolated from the mLN and LP of unchallenged *IL17a^{Cre} x R26R^{eYFP}* mice. Lines in the dot plots show mean values. Data are representative of n = 2 independent experiments.

Finally, we assessed mitochondrial content in TCRγδ positive and negative cells isolated from the IEL. Using Mitotracker Deep Red staining, we found that IEL TCRγδ⁻CD8αα⁺ cells, IEL TCRγδ⁻CD8αβ⁺ cells, and IEL TCRγδ⁺CD8αα⁺ cells all displayed a remarkably lower loading of Mitotracker Deep Red as compared to CD8αβ cell population isolated from the mLN (**Reviewer 6 – figure 8**).

Reviewer 6 – figure 8

Flow cytometry analysis of Mitotracker Deep Red staining in the indicated cell subsets isolated from the mLN and IEL of unchallenged GREAT mice. Lines in the dot plots show mean values. Data are representative of n = 1 experiment.

To further address the comment of Reviewer 6, we found that ongoing infection with *LmOVA* (day 4 and 10 post-infection) did not affect the PGE₂ concentration in the intestine (whole tissue), that remains

stable as compared to unchallenged mice, as well as considerably higher as compared to mLN and spleen (**Reviewer 6 – figure 9**).

Reviewer 6 – figure 9

Targeted metabolomics analysis of interstitial fluid isolated from spleen, mLN and small intestine of unchallenged C57BL/6J mice, and of mice challenged orally with *LmOVA*, 4 and 10 days after challenge. Dot plots show data from n = 1 experiment, with 2-3 biological replicates.

e) Finally, authors could have compared naïve OT-I T cells transferred to RAG-deficient recipient mice, which will migrate to the intestine as naïve cells but will not transition to effector cells or TRM cells, whereas feeding the recipients an OVA-diet will induce functional maturation of the OT-I cells in the tissue. Comparison of these two OT-I donor cell populations in the gut in terms of their metabolism/mitochondrial adaptation, would have been telling about the mechanisms of gut adaptation, with the same cells and the same environment but different Ag conditioning.

As previously stated, we are logistically unable to perform the suggested experiment. Moreover, as we stated above and in the previous rounds of answers to Reviewer 3, our paper does not aim at linking the reduction of mitochondrial content observed in gut-isolated cells to the acquisition of the T_{RM} fate (as also Reviewer 6 recognized at the beginning of his/her comments). However, we performed an alternative experiment to possibly address the comment of Reviewer 6, although not using an antigen-specific setting.

We assessed the mitochondrial content of naïve CD8⁺ T cells (T_N) isolated from mLN and Peyer's patches (PP), and compared them to IEL CD69⁺CD103⁺ cells. Our reasoning was that PP are intimately connected to the small intestine and are likely to be exposed to the same cues as compared to cells isolated from LP or IEL. We found that T_N isolated from PP slightly reduced their mitochondrial content as compared to mLN T_N, but not to the level observed in IEL CD69⁺CD103⁺ cells (**Reviewer 6 – figure 10**). While the environment of the small intestine may play a role in reducing the mitochondrial content of T_N in the PP, it is clear that T_N are less prone to the reduction of mitochondrial content as compared to effector T cells in the IEL. As we discussed in response to Reviewer 2, and as we stated in the discussion (lines 491-498), it is likely that the presence of PGE₂ in the gut microenvironment may not be sufficient to trigger the reduction of mitochondrial content. The expression of EP receptor is likely to

play a major role (see figure 3f and supplementary figure 5g), and we cannot exclude that the activation status of different cells subsets may influence the metabolic choices of cells isolated from LP and IEL of the small intestine.

Reviewer 6 – figure 10

Flow cytometry analysis of Mitotracker Green staining in the indicated cell subsets isolated from the mLN, PP and IEL of unchallenged C57BL/6J mice. Lines in the dot plots show mean values. Data are representative of n = 1 experiment. Statistical analysis was performed using one-way ANOVA.

Finally, to clearly state that our work does not address whether the acquisition of the tissue-resident memory phenotype by CD8⁺ T cells is linked to the reduction of mitochondrial content in the small intestine, we amended the manuscript discussion to highlight this limitation of our study (lines 498-503).

Comments on the metabolic aspects/data. Although not directly my expertise, I do have some comments and concerns

f) The finding that PGE2 participates in the metabolic regulation of the T cells as part of the adaptation to the gut environment is interesting and novel. However, the study is rather superficial. It is not clear how PGE2 is controlling the metabolic state of the T cells. It is also not clear why PGE2 would specifically act on the mucosal T cells. PGE2 effects are dose-dependent and different doses can have opposite effects.

Who is providing the PGE2? The activated T cells? The IECs? The PGE2 release has to be local since PGE2 is unstable. Is PGE2 also controlling the reversal state of reactivation in case of a secondary challenge? What controls that switch from inactive to active? Is it also driven by metabolic events?

In response to the criticism of Reviewers 2 and 4 we explored the mechanism underlying of how PGE₂ sensing regulates the mitochondrial content of T cells. Specifically, during the first round of revisions, we added a new figure 6 (now figure 5) to the manuscript, to show:

1. PGE₂ directly activates autophagy (figure 5a).

2. *Drp1* is not involved in the clearance of mitochondria and fitness of LP- and IEL-isolated cells (figure 5b-d).
3. The malate-aspartate shuttle (via *Got1*) mediates the PGE₂-driven changes in the mitochondrial membrane potential (figure 5e-j).

We believe that these data provided mechanistic insights on how PGE₂ regulates the mitochondrial content of CD8⁺ T cells isolated from LP and IEL of the small intestine, and strengthened our conclusion that the PGE₂-Got-1-autophagy axis is involved in controlling the metabolic fitness of CD8⁺ T cells in the intestine.

As stated in the discussion (lines 480-516), we think that PGE₂ acts on mucosal cells because it is a constitutive component of the intestinal barrier (Miyoshi H, EMBO J 2017; Patankar JV, Nat Cell Biol 2018), and because CD8⁺ T cells in the mucosa express the receptors to sense it. We think that this is not exclusive to the intestine, as other microenvironments have been described to be enriched in PGE₂ (such as the tumor microenvironment, Zelenay S, Cell 2015) and it is possible that as long as cells express EP receptors, their mitochondrial content may be affected by PGE₂. Indeed, PGE₂ can trigger reduction of mitochondrial content in CD8⁺ T cells, *in vitro* (figure 3e). While we used IL-15 and TGF-β to “mimic” T cell exposure to the intestinal environment, we are well aware that *in vitro* cells cannot resemble the *in vivo* phenotype. However, these data support the hypothesis that the PGE₂ influence on the mitochondrial content of CD8⁺ T cells may not be limited to the gut environment.

Regarding the comment of Reviewer 6 on the opposite effects of different doses of PGE₂, we did not observe this phenomenon in our system. As shown in supplementary figure 5f, doses of PGE₂ between 10 nM and 10 μM triggered similar responses, *ie* the reduction of mitochondrial content and function. On the other hand, a dose of 1 nM failed to trigger a response.

We sorted different CD8⁺ T cell subsets from unchallenged C57BL/6J mice and tested the activity of phospholipase A₂ (PLA₂), essential for the processing of arachidonic acid towards generation of PGE₂, but we failed to detect any PLA₂ activity in our T cell samples.

It is likely that major producers of PGE₂ in the small intestine LP are non-bone marrow-derived stromal cells (Newberry RD, J Immunol 2001) and mesenchymal stem cells (Manieri NA, Gastroenterology 2012).

We did not test whether PGE₂ was controlling the reactivation of T cells in the gut, as we did not assess memory time points. However, we want to point out to the attention of Reviewer 6 that CD8⁺ CD62L⁻ CD44⁺CD69⁺CD103⁺ T cells in the mLN, defined to be, at least in an antigen-specific setting, gut-derived T_{RM} that re-entered the systemic circulation (Beura LK, Immunity 2018), have mitochondrial content comparable to other mLN-isolated subsets. This suggests that the mitochondrial phenotype of CD8⁺ T cells is flexibly regulated, and likely dependent on the exposure to PGE₂, as mLN have substantially lower levels of PGE₂. Of course, this speculation must be confirmed in an antigen-specific setting, as suggested in the discussion.

As suggested by Reviewer 6, it is likely that the reactivation of LP and IEL CD8⁺ T cells depends on metabolic events, to support their proliferation and effector functions during re-challenge. Recently, work from the group of Marc Veldhoen highlighted how activation of intestinal tissue-resident cells depends on metabolite availability and engagement of specific metabolic pathways (Konjar S, PNAS USA 2022).

g) I agree with reviewer #3, that the in vitro system with IL15/TGF β is not appropriately reflecting the in vivo situation in the gut. Moreover, PGE₂ has significant effects on TGF β signaling. Are both needed in vivo for the effect on the T cell metabolism in the gut? If so it could explain, why PGE₂ has such specific effect in the gut, where TGF β is abundantly expressed and central for controlling several gut adaptation processes.

The hypothesis of Reviewer 6 is intriguing. While the in vitro system we used did not reflect the in vivo situation in the gut, as we state in the answer to Reviewer 3 and in the manuscript, we think this system is well suited to address, at least in part, the point on Reviewer 6. We set up a CD8⁺ T cell culture as described in the paper, and treated the cells with IL-2, IL-15, and IL-15 + TGF- β . On the last 24h of the cultures we treated cells with or without PGE₂. As suggested by Reviewer 6, we found that PGE₂ treatment impinged on the expression of CD103, the expression of which is well known to be controlled by TGF- β . Interestingly, PGE₂ was able to drive reduction of the mitochondrial content upon IL-15 + TGF- β and IL-15 alone conditions, suggesting that TGF- β signaling, at least in our setting, is not required to mediate the effect of PGE₂ on the metabolic fitness of CD8⁺ T cells (**Reviewer 6 – figure 11**).

If Reviewer 6 deems appropriate, we are happy to include these data in the manuscript and to discuss these findings.

Reviewer 6 – figure 11

Flow cytometry analysis of CD103 and Mitotracker green staining in CD8⁺ T cells activated in IL-2-, IL-15- or IL-15/TGF- β -polarizing conditions for 6 days and treated for 24 hours with 100 nM PGE₂. Lines in the dot plot show mean values and the data are representative of n = 2 independent experiments. Statistics were performed using two-way ANOVA and Sidak's multiple comparison correction.

Unfortunately, this was not addressed or even considered in this paper. Overall, although the PGE2 aspect is new and interesting, it is poorly developed and in agreement with reviewer 3, this study leaves the reader with many more questions than answers.

REVIEWERS' COMMENTS

Reviewer #6 (Remarks to the Author):

The authors made a significant effort to address the additional reviewers comments/suggestions and concerns. Although there are still some glitches (no paper is perfect!), in my opinion, they are not significant enough to prevent the manuscript from moving forward.

The new experiments (Fig 11) and findings included in the latest rebuttal to address the role of TGFb and IL15 are important (and perhaps surprising) and it would indeed be advancing to include and discuss these in light of existing knowledge in the final version of the manuscript.

Point-by-point response – 28.11.2023

Nature Communications manuscript NCOMMS-23-11142A

Reviewer #6 (Remarks to the Author):

The authors made a significant effort to address the additional reviewers comments/suggestions and concerns. Although there are still some glitches (no paper is perfect!), in my opinion, they are not significant enough to prevent the manuscript from moving forward.

We thank Reviewer 6 for acknowledging our work and efforts.

The new experiments (Fig 11) and findings included in the latest rebuttal to address the role of TGFb and IL15 are important (and perhaps surprising) and it would indeed be advancing to include and discuss these in light of existing knowledge in the final version of the manuscript.

We followed the Reviewer's suggestion and implemented the findings showed in Reviewer Figure 11 as a new panel in the Supplementary Figures file (Supp. Fig. 5f). We report and discuss these findings at lines 275-278 of the revised manuscript.